# Precision Nutrition and the Microbiome, Part I: Current State of the Science

**DOI:** 10.3390/nu11040923

**Published:** 2019-04-24

**Authors:** Susan Mills, Catherine Stanton, Jonathan A. Lane, Graeme J. Smith, R. Paul Ross

**Affiliations:** 1APC Microbiome Ireland, University College Cork, Cork T12 K8AF, Ireland; susan.mills@ucc.ie; 2APC Microbiome Ireland, Teagasc Food Research Centre, Fermoy P61 C996, Co Cork, Ireland; catherine.stanton@teagasc.ie; 3H&H Group, Technical Centre, Global Research and Technology Centre, Cork P61 C996, Ireland; jonathan@hh.global

**Keywords:** personalised nutrition, precision nutrition, probiotics, prebiotics, gut microbiome, immunity, metabolic disease, gut, genetics

## Abstract

The gut microbiota is a highly complex community which evolves and adapts to its host over a lifetime. It has been described as a virtual organ owing to the myriad of functions it performs, including the production of bioactive metabolites, regulation of immunity, energy homeostasis and protection against pathogens. These activities are dependent on the quantity and quality of the microbiota alongside its metabolic potential, which are dictated by a number of factors, including diet and host genetics. In this regard, the gut microbiome is malleable and varies significantly from host to host. These two features render the gut microbiome a candidate ‘organ’ for the possibility of precision microbiomics—the use of the gut microbiome as a biomarker to predict responsiveness to specific dietary constituents to generate precision diets and interventions for optimal health. With this in mind, this two-part review investigates the current state of the science in terms of the influence of diet and specific dietary components on the gut microbiota and subsequent consequences for health status, along with opportunities to modulate the microbiota for improved health and the potential of the microbiome as a biomarker to predict responsiveness to dietary components. In particular, in Part I, we examine the development of the microbiota from birth and its role in health. We investigate the consequences of poor-quality diet in relation to infection and inflammation and discuss diet-derived microbial metabolites which negatively impact health. We look at the role of diet in shaping the microbiome and the influence of specific dietary components, namely protein, fat and carbohydrates, on gut microbiota composition.

## 1. Introduction

The human gastrointestinal tract (GIT) is considered one of the most densely populated ecosystems on our planet, purported to harbour ~10^13^ microorganisms [1] referred to as the gut microbiota, whose activities have significant consequences for the host in terms of health and disease. This is unsurprising given that the entire genetic content of the human gut microbiota, commonly referred to as the gut microbiome, is estimated to exceed human genomic content by a factor of ≥100 [2], with vast genetic potential to contribute to host physiology. It is composed of the three domains of life, Bacteria, Archaea, and Eukarya (fungi, protozoans and metazoan parasites), as well as eukaryotic and prokaryotic viruses (bacteriophages). 

The study of the gut microbiome has been revolutionised over the last 15 years by advances in genetic methods and sophisticated bioinformatic tools. Next generation sequencing is a low-cost, high-throughput sequencing platform that enables analysis of all the genomes within an ecosystem sample (shotgun metagenomics), or a description of the taxa within a given community by sequencing conserved marker genes, such as the 16srRNA gene of bacteria and archaea (marker gene metagenomics) thus removing the requirement for cultivation of clonal cultures [3]. Indeed, such is the interest in the gut microbiota, it has been reported that in 2017 alone, approximately 4000 research papers on the topic were published, with more than 12,900 papers published in the preceding four years [4], and to date the bacterial component has received the most attention. This vast array of data has contributed to a deeper understanding of the role of the gut microbiome in health and disease, the factors which shape it, and the potential to harness its therapeutic potential for optimal health. 

The gut microbiota serves the host by interacting directly or indirectly with host cells, the latter of which comes about through microbially-produced bioactive molecules, thus the microbiota is capable of regulating numerous biological pathways involved in immunity and energy homeostasis, while also protecting the host against pathogens through colonisation resistance. The dysbiotic microbiota, which has deviated from the ‘healthy’ status in terms of diversity and functionality, has been implicated in a range of diseases, including inflammatory bowel diseases (IBDs) [5], cancer [6], neuropsychiatric disorders [7] and cardiometabolic diseases, including obesity, type 2 diabetes and cardiovascular disease [8]. These diseases tend to inflict those living a Westernised lifestyle. Indeed, greater economic development has been shown to correlate with significantly lower within-host gut microbiota diversity, a common feature of the dysbiotic microbiota [9]. 

Many researchers have suggested that the gut microbiome can be considered as a virtual organ but unlike any other organ in the human body, the gut microbiome represents a source of significant inter-individual variation, rendering the analysis of it ever more complicated. However, this vast inter-individual variation is now being seized upon as an opportunity to utilize the microbiome for precision medicine [10,11] and personalised nutrition [12]. Indeed, along with the human genome, the human microbiome has been implicated as the main source of human variation modulating dietary responses [12]. This variation presumably goes a long way to explaining the pandemic surge in metabolic diseases despite the fact that universal advice and education on healthy eating practices have never been more readily available. 

Therefore, the overall aim of this two-part review is to investigate the potential of the microbiome for precision microbiomics, particularly in relation to nutrition. In this regard, precision microbiomics can be described as the use of the gut microbiome as a biomarker to predict the effect of specific dietary components on host health and the use of these data to design precision diets and interventions that ensure optimal health. 

In Part I, we describe the development of the gut microbiota throughout life along with the factors that shape its composition and how it contributes to host health. We look at dietary practices and specific dietary components that influence the composition and functionality of the gut microbiota and the consequences for human health with a focus on infection, inflammation and metabolic disease. In Part II, we focus on the potential use of the microbiome to prevent disease and promote health. We present research regarding opportunities to modulate the microbiota for prevention of over- and under- nutrition through probiotics, prebiotics and fibre and we look at the impact of environment and life-stage on the gut microbiota and health and dietary strategies and interventions for optimising health. Finally, in Part II, we provide evidence from recent research that the gut microbiome has potential to serve as a biomarker of human responsiveness to diet and assess where this current knowledge-base places us in terms of putting precision microbiomics into everyday practice and discuss the value of present-day commercial microbiome testing.

## 2. Development of Microbiota from Birth/Young versus Mature versus Aged Microbiota

Humans encounter their initial gut microbiota in very early life with most studies focusing on colonisation after birth and the microbial assembly that ensues. Some studies have led to the hypothesis that colonisation begins in utero by providing evidence of a placental microbiome in healthy pregnancies, the presence of microorganisms in the amniotic fluid and meconium. These studies have been presented and critically assessed in a recent review by Perez-Muňoz et al. [13]. The authors concluded that there is insufficient evidence to date to support the ‘in utero colonisation hypothesis’, since the studies lacked appropriate controls for contamination, the molecular approaches used were incapable of studying low-biomass microbial populations and evidence of bacterial viability was not provided. Thus, for the purpose of this review, we have focused on gut microbiota development from birth onwards. 

Although gut microbiota composition varies considerably between infants certain patterns have been identified. Several factors have been shown to influence microbial colonisation of the gut at this stage, including gestational age, birth mode, sanitation, antibiotic exposure, feeding regime and host genetics [14,15,16,17,18,19,20]. However, facultative anaerobes have been identified as the initial colonisers [21]. Depletion of the available oxygen creates the necessary environment for the establishment of strict anaerobes [16,21]. Anaerobes which colonise the gut in the early days and weeks of life include *Bifidobacterium, Bacteroides, Clostridia* and *Parabacteroides* [16,18,21,22]. Although the diversity of the microbiota is generally low at this stage, dominated largely by members of the Actinobacteria phylum in the case of full-term spontaneously vaginally delivered infants, it has been shown to be greatest in this cohort compared to full-term infants delivered by caesarean section (dominated by Firmicutes) or pre-term infants (dominated by Proteobacteria) at one week old [16]. Interestingly, by week 24, no significant differences in alpha diversity were recorded between any of these groups [16]). Diversity increases with age with a gradual increase in the presence of Firmicutes and Bacteroidetes by the first year [17] (Canadian infants); [20] (Canadian infants); [18] (Swedish infants) where the introduction of solid food to the diet has been identified as an important step in the succession of the microbiota [22] (Spanish infants), and [23] (Danish infants). In terms of metabolic function, genes involved in the *de novo* biosynthesis of folate have been shown to be enriched in the infant microbiome across three different populations (Malawian rural communities, Amerindians from the Amazonas of Venezuela, families from the USA) relative to adults [24]. By three years of age, obligate anaerobes have been shown to dominate the microbiota in breast-fed infants [21] which is trending towards an adult-like composition. The establishment of a stable adult-like microbiota occurs between 2–5 years of age and is dominated by Firmicutes and Bacteroidetes [25,26,27]. 

Few studies have specifically investigated the pre- adolescent and adolescent microbiota of healthy humans. However, those which have investigated these age groups indicate that the microbiota has not yet reached the adult-state and is providing essential functions towards the developmental process of its human host. The gut microbiota of children is more stable than that of infants with composition largely influenced by dietary habits and geography [28]. The pre-adolescent microbiota (7–12 years of age) is still in a state of immaturity and based on observations from a group of children from Houston, Texas, has been shown to be more diverse and harbor significantly greater abundances of Firmicutes and Actinobacteria than observed in healthy adults [26]. The pre-adolescent microbiome was also found to be enriched in functions potentially involved in ongoing development, such as vitamin B12 synthesis and de novo folate synthesis relative to the adult microbiome. In terms of the adolescent microbiota, Agans et al. [29] identified a core microbiota of 46 species common to both adults and adolescents (11–18 years) who consumed a standard Western diet, however, the abundances of the genera *Bifidobacterium* and *Clostridium* were significantly higher in adolescents relative to adults. 

The healthy adult gut microbiota is composed primarily of the phyla Firmicutes and Bacteroidetes, and to a lesser extent the phyla Actinobacteria, Proteobacteria, and Verrucomicrobia [30,31]. As with any age-group, due to the extensive inter-individual variation, it has been virtually impossible to define the composition of the ‘healthy’ adult gut microbiota. However, the ‘enterotype’ concept was introduced in 2011 [32]) when faecal metagenomes of individuals from America, Europe and Japan were found to be dominated by one of three different bacterial communities, namely *Bacteroides* (enterotype 1), *Prevotella* (enterotype 2) or *Ruminococcus* (enterotype 3). The enterotype ‘concept’ has since been used in other studies when evaluating the gut microbiota as we will see further on, although further analysis has resulted in the identification of only two enterotypes, one of which is dominated by *Prevotella* and the other by *Bacteroides* which have been linked to long term carbohydrate-, or animal fat and protein-rich diets, respectively [33]. More recently, it has been suggested that *Prevotella* and *Bacteroides* should be interpreted as ‘biomarkers’ of diet, lifestyle and disease state given that gradients of both *Bacteroides* and *Prevotella* have been found within gut communities as opposed to distinct and consistent community taxa [34]. 

Reports on the estimated number of species and strains in an individual have varied greatly but in a study examining the stability of the gut microbiota in 37 US adults over a five year period, Faith et al. [35] reported an average of 101 ± 27 species and 195 ± 48 strains in the faecal gut microbiota of each individual with family members sharing strains, which was not observed in unrelated individuals. However, other studies estimate the number of species to be greater than 1000 [36,37]. More recently, Forster et al. [38] presented the Human Gastrointestinal Bacteria Culture Collection which consisted of 737 whole-genome-sequenced bacterial isolates, representing 273 different species, including 105 novel species from 31 families found in the human gastrointestinal microbiota. The healthy adult microbiota has been shown to be stable over long periods of time [35] but can be influenced by a number of factors. These include geographical location [24,39] albeit diet would appear to be an important contributing factor in this regard [40,41,42], direct antibiotic usage [43], and indirectly by consumption of antibiotic containing animal derived products, such as beef and chicken as a result of their use in livestock production [44], non-antibiotic drugs [45], illness, injury [46,47] and hormonal changes [48]. The healthy gut microbiota is generally characterised by rich species diversity [49] which has been found to be reduced/altered in individuals with certain diseases, particularly those typified by a dysregulated immune response (as discussed further on). 

Ageing has a significant impact on the gut microbiota with dramatic compositional and functional changes observed in the elderly microbiota (in general >65 years). Several physiological and lifestyle changes associated with the ageing process may be contributing factors resulting in changes in dietary habits and ultimately nutrition, including a decline in dentition and salivary function, a reduction in digestion and absorption, due to gastrointestinal dysmotility, changes in appetite as a result of prescribed drugs and psychological state, or changes in living conditions, such as residential care or hospitalisation [50]. Gastric hypochlorhydria which is associated with ageing and is prevalent in individuals experiencing or who have experienced *Helicobacter pylori* infection can cause malabsorption and bacterial overgrowth in the small intestine [50,51]. 

In general, the elderly microbiota has been characterised by a decline in microbial diversity, an increase in the abundance of opportunistic pathogens and a decrease in species associated with short chain fatty acid (SCFA) production, in particular butyrate [52]. The inter-individual variation observed in the gut microbiota of adults is even greater in the elderly cohort. Indeed, while Bacteroidetes was found to be the dominant phylum in an elderly Irish cohort (age 65 years and older), the proportion of Bacteroidetes within individual composition datasets ranged from 3% to 92%, while the proportion of Firmicutes ranged from 7% to 94% [53]. Mariat et al. [54] reported a change in the ratio of Firmicutes: Bacteriodetes with age, which was recorded as 0.4 in infants (3 weeks to 10 months old), 10.9 in adults (25–45 years old) and 0.6 in the elderly cohort (70-90 years), of which the latter two consumed an unrestricted Western-type diet. In a study involving 178 elderly Irish subjects (65–96 years), Claesson et al. [55] identified distinct microbiota composition groups as a result of residence location (community versus day-hospital versus rehabilitation versus long-stay residential care) which also overlapped with diet (low fat/high fibre versus moderate fat/high fibre versus moderate fat/low fibre versus high fat/low fibre, respectively). The gut microbiota of people in long-stay care was found to be significantly less diverse than that of healthy community dwellers, which was found to be more similar to healthy young adults. Furthermore, increased frailty as observed in the less healthy, long-stay subjects correlated with a loss of community-associated microbiota. Indeed, a distinct negative correlation between frailty and gut microbiota alpha diversity has been reported [56]. In the same study, the species *Eubacterium dolichum* and *Eggerthella lenta* were found to be more abundant with frailty, while a *Faecalibacterium prausnitzii* operational taxonomic unit (OTU) was less abundant in frailer individuals. Interestingly, *F. prausnitzii* is an important butyrate producer [57], while *E. lenta* is considered a pathogen [58]. More recently, Haran et al. [59] also reported lower abundances of butyrate-producing organisms in the microbiota of American elderly nursing home residents (age 65 years or older) consuming a low-fibre, typical nursing home diet with increasing frailty and higher abundances of recognised dysbiotic species. Increasing age was also associated with a decrease in the abundance of microbiome-encoded genes and pathways associated with vitamin B, a nitrogenous base and essential amino acid production. 

Interestingly, the centenarian microbiota (99–104 years) has been shown to significantly differ from that of young adults (25–40 years) and even elderly subjects (63–76 years) [60] (Italian subjects), being characterised by an enrichment in facultative anaerobes, mainly pathobionts, and a rearrangement in the Firmicutes population with a marked decrease in symbiotic species associated with anti-inflammatory properties e.g., *F. prausnitzii* and relatives. Increased inflammatory markers were also identified in centenarian blood samples, indicative of increased inflammatory status. Rampelli et al. [61] also reported a rearrangement in Firmicutes in Italian centenarians (99–102 years), which was not observed in 70-year-old subjects. In this study a functional description of the coding capacity of an ageing human cohort revealed a decrease in saccharolytic potential and an increased abundance of proteolytic functions in the centenarian microbiome. Remarkably, a study investigating the semi-supercentarian microbiota (105–109 years old) revealed not only a decrease in saccharolytic butyrate producers (*Faecalibacterium*, *Coprococcus*, *Roseburia*) and an increase in potential opportunistic bacteria, as expected, but also an enrichment and/or higher prevalence in health associated bacterial groups, including *Akkermansia*, *Bifidobacterium* and *Christensenellaceae*. Whether these species were present at a younger age or related to past lifestyle or genetics is not known [62]. 

In certain regions of the world, “healthy” ageing is the norm and the average lifespan of residents surpasses general averages. Goatian village in Liuyang city of Hunan province in mid-South China is such an example, which boasts an average lifespan of 92, much higher than the average 74.83 years for China in general, along with a lack of chronic diseases amongst its people. An analysis of the gut microbiota in its long-living elderly residents (ages spanning from 50 to >90 years) revealed much greater species diversity than that observed in the control group (healthy subjects from other areas in China, average age of 50 years) [63]. Similarly, Bian et al. [64] reported that the overall gut microbiota composition of healthy aged Chinese subjects was similar to that of much younger people, reporting little differences between individuals from 30 to >100 years which the authors speculate may be a consequence of the healthy lifestyle and diet of the study subjects. 

## 3. Role of Microbiome in Health, Development and Immune Functioning

The gut microbiome is integral to the health of its host, serving a myriad of functions. It provides essential nutrients and bioactive metabolites, which can be produced directly by the microorganisms or indirectly by microbial conversion of host or environmental molecules. It is involved in energy regulation (as discussed in Section 5). It prevents pathogen colonisation directly or indirectly through a phenomenon referred to as colonisation resistance. It sustains the integrity of the mucosal barrier and is an essential component in the orchestration of immune functioning within the gut. The bidirectional interactions within the brain-gut-microbiome axis, in which the gut microbes communicate to the central nervous system, have been demonstrated largely by preclinical and some clinical studies [65]. Alterations in brain-gut-microbiome communication have been implicated in various disease states, from irritable bowel syndrome to psychiatric and neurologic disorders, and this is an area of research that has the potential for identification of novel therapeutic targets and therapies [66]

### 3.1. Nutrients and Bioactive Metabolites

Fruits, vegetables and cereals are major components of the human diet providing essential carbohydrates and dietary fibres, although digestion of the latter is beyond the scope of the human genome [67]. Cantarel et al. [68] identified only 17 enzymes within the human genome to breakdown carbohydrate nutrients which included starch, lactose and sucrose. Thus, plant cell wall polysaccharides and resistant starch, which constitute most dietary fibres and cannot be digested or absorbed in the small intestine, enter the large intestine and undergo microbial breakdown and subsequent fermentation [67]. The microbiota also feeds on animal-derived dietary carbohydrates (glycosaminoglycans and N-linked glycans from cartilage and tissue), host epithelial glycome, and microbe-derived carbohydrates from resident gut microbes or foodborne microbes [69,70]. Collectively, the carbohydrates consumed by the microbiota have been termed “microbiota accessible carbohydrates” (MACs) [70]. 

Carbohydrate active enzymes (CAZymes) breakdown MACs into fermentable monosaccharides [67]. For example, the gut bacterium *Bacteroides thetaiotaomicron* was recently shown to metabolise the most structurally complex plant polysaccharide known, rhamnogalacturonan-II, using a highly specific enzyme system [71]. *Bifidobacterium longum* strains derived from the infant gut were shown to be capable of metabolising human milk oligosaccharides [72]. In-silico analysis of CAZyme profiles in the guts of 448 individuals from diverse geographies and age groups revealed 89 CAZyme families which were present across 85% of the gut microbiome and revealed several geography/age-specific trends in the CAZyme repertoires of individuals [73]. The major end products of microbial fermentation of the resulting monosaccharides are the SCFAs, including butyrate, propionate, and acetate which reach a combined concentration of 50–150 mM in the colon at a ratio of 1:1:3, respectively [74]. They are rapidly absorbed by the intestinal epithelial cells where they are involved in a number of cellular and regulatory processes [75,76] with only 5% excreted in faeces [77]. Butyrate is mainly produced by Firmicutes, propionate by Bacteroidetes, and acetate by most gut anaerobes [78]. Butyrate is the main energy source for the epithelial cells [79] and plays an important role in brain function [80]. It is also known for its anti-cancer [81,82,83] and anti-inflammatory properties [77,84] and for its role in the development of the intestinal barrier [85,86,87]. Propionate contributes to gluconeogenesis in the liver [86] and, along with butyrate, has been shown to activate intestinal gluconeogenesis, albeit both use different circuits [88]. Propionate derived from the gut microbiota has also been shown to reduce cancer cell proliferation in the liver [89]. The SCFAs are also involved in regulating immune responses, a topic which has been extensively reviewed by Corrêa et al. [90]. For example, acetate has recently been shown to promote intestinal antibody IgA responses to the gut microbiota via the G protein coupled receptor GPR43 [91]. Intestinal IgA is specialised in protecting the mucosa [92]. These SCFAs also stimulate secretion of gut hormones, such as glucagon-like peptide 1 (GLP-1) and plasma peptide YY (PYY) involved in appetite regulation and satiety from enteroendocrine cells [93,94] proposedly through the SCFA receptors GPR41 and GPR43 [95,96], thus playing a role in energy regulation in the body. Unsurprisingly, changes in the production of these compounds as a result of disturbances to the gut microbiota can result in pathological consequences for the host. As an example, increased acetate production from an altered gut microbiota in a rodent model was shown to promote metabolic syndrome [97]. 

The gut microbiota is also responsible for the biosynthesis of several essential vitamins, including B vitamins, such as cobalamin, folic acid, biotin, thiamine, riboflavin, nicotinic acid, pyrodixine and pantothenic acid, as well as vitamin K [98]. Interestingly, Arumugam et al. [32] observed vitamin biosynthesis pathways across the three identified enterotypes; however, enterotype 1 was enriched in the biosynthesis of riboflavin, biotin, ascorbate and pantothenic acid, while enterotype 2 was enriched in the biosynthesis of thiamine and folic acid. 

Primary bile acids are produced in the liver from dietary cholesterol and cholesterol derived from hepatic synthesis, and their main function is to aid absorption of dietary lipids and lipid soluble nutrients [99]. However, bile acids are also important signalling molecules and are known to activate a number of nuclear receptors, including farnesoid X receptor (FXR), preganane X receptor, and vitamin D receptor, as well as the G-protein-coupled receptor TGR5, and cell signalling pathways in the liver and GIT thus modulating their own biosynthesis, as well as glucose, lipid, and energy metabolism [100]. In humans, 200-800 mg of bile acids escape enterohepatic circulation every day, pass into the colon and are metabolised by bacteria to secondary bile acids [99]. In a mouse model, such secondary bile acids produced through the action of bacterial bile salt hydrolase (BSH) have been shown to regulate weight gain, lipid metabolism and cholesterol levels via regulation of key genes in the liver or small intestine [101]. The gut microbiota has also been shown to inhibit bile acid synthesis in the liver by alleviation of FXR inhibition in the ileum [102]. 

In recent years, there has been a growing appreciation for the ability of the gut microbiota to produce neurochemicals that can influence the peripheral enteric and central nervous systems [103]. For example, gamma amino butyric acid (GABA) is a major inhibitory neurotransmitter in the brain [104] and neuropsychiatric disorders, including anxiety and depression have been linked to GABA system dysfunction [105]. Strains of culturable lactobacilli and bifidobacteria from the human intestine were shown to produce GABA, namely *Lactobacillus brevis*, *Bifidobacterium dentium*, *adolescentis* and *infantis* [106]. Furthermore, GABA was found to serve as a growth factor for a previously uncultured gut bacterium, *Flavonifractor sp.* which was shown to ferment GABA [107]. In the same study, the authors identified several gut bacteria capable of producing GABA which included *Bacteroides*, *Dorea*, *Parabacteroides*, *Alistipes* and *Ruminococcus* species. More recently, a co-culture experiment revealed that GABA produced by *Bacteroides fragilis* was essential for the growth of a gut isolate termed KLE1738 which is believed to be an unreported bacterial genus [108]. This led to the isolation of a variety of GABA-producing bacteria and the *Bacteriodes* species in particular were found to produce large quantities of GABA. Furthermore, in the same study relative abundance levels of faecal *Bacteriodes* negatively correlated with brain signatures associated with depression in patients with major depressive disorder. Bacterially-produced GABA is generated by the enzyme glutamate decarboxylase (GAD) which catalyzes the irreversible α-decarboxylation of glutamate to GABA and is believed to protect the microorganism against stomach acidity [109]. GAD is also found in higher plants and animals [110]. Interestingly, daily consumption of a GABA-producing *Bif. dentium* gut isolate was found to modulate sensory neuron activity in a rat faecal retention model of visceral hypersensitivity revealing that bacterially-produced GABA can modulate abdominal pain [111]. GABA-enriched black soybean milk fermented with a GABA-producing fish gut isolate, *L. brevis*, generated similar antidepressant activity in rats as the common antidepressant drug, fluoxetine, but without the side effects normally associated with the drug, such as appetite loss and reduced body weight [112]. 

Serotonin (5-hydroxytryptamine, 5-HT) is a brain neurotransmitter and performs regulatory functions in the gut and other organ systems [113]. It is derived from the amino acid tryptophan and plays an important role in the regulation of mood [114] such that several antidepressants act on serotonin transporters in the brain [115]. Yano et al. [113] demonstrated that human- and mouse-derived gut bacteria promote serotonin biosynthesis in colonic enterochromaffin cells which supply serotonin to the lumen, mucosa and circulating platelets. Spore forming bacteria, dominated by clostridial species, were found to elicit this effect. Furthermore, conventional mice were found to have 2.8 times more plasma serotonin levels than their germ-free counterparts [116]. This peripherally produced molecule does not pass the blood brain barrier under physiological conditions [117] but is an important signalling molecule in the gut involved in peristalsis, secretion, vasodilation, pain perception and nausea, as well as promoting inflammation and being involved in neuron development and maintenance in the enteric nervous system, while platelet serotonin derived from the gut influences bone development amongst other functions [118,119]. The mechanisms involved in gut microbiota-mediated serotonin biosynthesis have not yet been fully elucidated but are thought to be linked to microbial metabolite-stimulation of the enzyme tryptophan hydroxylase 1 in enterochromaffin cells which produces a serotonin precursor that is subsequently metabolised to serotonin [113]. In particular, rectal injection with the microbial metabolites deoxycholate, p-aminobenzoate, α-tocopherol and tyramine, were shown to increase colonic and serum concentrations of serotonin in mice [113].

Metabolism of tryptophan in the gut results in tryptophan catabolites which have profound effects on the host [120]. Direct transformation of tryptophan by intestinal microbes results in the formation of several molecules, including ligands for the ligand-activated aryl hydrocarbon receptors (AhRs) [117]. AhRs, which are transcription factors, are expressed by several cells of both the adaptive and innate immune systems [121]. Upon binding of AhR to its ligand molecule, the activated transcription factor translocates into the nucleus of the cell where it mediates cell-specific transcriptome changes [122]. Thus, AhR signalling plays a key role in immune functioning in health and disease. Lamas et al. [123] showed that intestinal inflammation in mice harbouring a microbiota incapable of metabolizing tryptophan was attenuated following treatment with *Lactobacillus* strains capable of activating AhRs through the production of tryptophan metabolites. In the same study, faecal samples from healthy subjects induced significantly greater AhR activation compared to faecal samples from IBD subjects, the latter of which harbored significantly less tryptophan and tryptophan metabolites. The tryptophan-metabolising strain *Peptostreptococcus russellii* provided a protective effect against colitis in mice, which the authors suggest is linked to its ability to produce the tryptophan metabolite and AhR ligand, indoleacrylic acid which mitigates inflammatory responses and promotes intestinal barrier function [124]. The authors also noted a diminishment of tryptophan-metabolising capacity in stool samples of IBD patients. The metabolite indole has been shown to enhance epithelial barrier function and attenuate indicators of inflammation [125,126].

### 3.2. Colonisation Resistance

The intestinal microbiota protects its host against colonisation by exogenous pathogens and prevents the overgrowth of potentially pathogenic endogenous members, referred to as colonisation resistance [127]. This phenomenon is elicited through competition for nutrients and colonisation sites, direct inhibition of pathogens through the production of antimicrobial substances, and indirectly through modulation of the luminal environment and via host-commensal interactions involving the epithelial barrier function, modulation of the host cell surface and the host immune system [128]. 

Members of the established microbiota are controlled by ‘substrate competition,’ defined as the superior ability of a species/strain to utilise one or a few substrates over other species and the control of that population by the limited concentration of these substrates [129]. Furthermore, one microorganism’s by-product can serve as substrate for another [127]. In this regard, nutrient resources in the gut are in huge demand and simultaneously limited, making it challenging to become established or outcompete with resident microbiota. Indeed, nutrient utilisation by the colonic microbiota of mice was shown to restrain the growth of *Clostridium difficile,* since it was incapable of competing with the mouse microbiota for the available carbon sources [130]. Unfavourable environmental conditions created as a result of commensal fermentations can also inhibit the growth of undesirable microorganisms. The utilisation of human milk oligosaccharides, particularly the dominant secretor associated oligosaccharide 2′-fucosyllactose, by infant strains of bifidobacteria led to an increase in their proportions, an increase in lactate concentration and a subsequent reduction in pH which was shown to decrease the proportions of *Escherichia coli* and *Clostridium perfringens* during *in vitro* anaerobic fermentations [131]. Furthermore, consumption of butyrate by epithelial cells as an energy source has been implicated as important in maintaining a hypoxic environment in the gut lumen [132]. Decreased intestinal butyrate levels due to depletion of a commensal butyrate producer in a mouse model resulted in increased epithelial oxygenation and aerobic expansion of *Salmonella enterica* serovar Typhimurium [133]. Butyrate has also been shown to down-regulate virulence gene expression in *Salmonella* [134].

In terms of niche competition, Lee et al. [135] identified specific colonisation factors conserved within the *Bacteroides* genus, one of the most prominent genera of the human microbiota. The specific genetic locus referred to as *c*ommensal *c*olonisation *f*actors (*ccf*) was shown to be up-regulated in *Bac. fragilis* during gut colonisation, especially at the colonic surface such that the strain was able to reside deep within the crypt channels whereas *ccf* mutants were defective in terms of crypt association. Such species-specific physical interactions with the host provide an example of direct competition for niche occupation. 

The gut microbiota is a rich reservoir of bacteriocin producers [136,137]. Bacteriocins are ribosomally synthesised peptides with antimicrobial activity against either a broad range of species or a narrow range of closely-related species. Their mode of action varies depending on the bacteriocin class, but they generally exert their antimicrobial activity by forming pores in the target cell (Classes I and II), by degrading cell wall peptidoglycan (Class III bacteriolysins) or interfering with cellular processes (Class III non-lytic bacteriocins) [138]. The genetic machinery for bacteriocin synthesis is encoded in gene clusters or operons where many of the genes are conserved. Based on this knowledge, Walsh et al. [139] identified 74 bacteriocin gene clusters within the genomes of the GIT subset of the Human Microbiome Project’s reference genome database using an *in-silico* approach of which the most commonly identified were bacteriolysins, then lantibiotics and sactibiotics. Thuricin CD is an example of a sactibiotic bacteriocin produced by a human gut isolate, *Bacillus thuringiensis* [140]. While it has a narrow spectrum of inhibition, thuricin CD is capable of killing a wide range of *C. difficile* isolates, its antimicrobial activity being as potent as the antibiotics vancomycin and metronidazole, but without the concomitant damage to other members of the microbiota [141]. Bacteriocin production can also aid niche occupation of the producing strain. Indeed, bacteriocin production in *Enterococcus faecalis* harboring the bacteriocin-encoding conjugative plasmid pPD1 was shown to replace indigenous enterococci and out-competed *E. faecalis* strains lacking the plasmid in a mouse model [142]. Supplementing mice with bacteriocin-producing strains resulted in transient advantageuous changes, such as inhibition of *Staphylococcus* by enterocins and *Enterococcus* by garvicin and promotion of LAB by sakacin, plantaricins and garvicin [143]. 

Other antimicrobials generated by the gut microbiota can also aid colonisation resistance. For example, a single bacterial species, namely *Clostridium scindens*, was shown to confer colonisation resistance against *C. difficile* infection *in vivo* [144]. In this instance, secondary bile acids generated by *C. scindens* from host-derived bile cells were found to inhibit the pathogen. 

### 3.3. Immunity and Mucosal Integrity

Commensal interactions with the host promote immune system maturation and immune homeostasis through complex microbiota-host networks although much of our understanding of these mechanisms today have been extrapolated from animal or *in vitro* studies [145,146,147]. We have already seen how several microbial metabolites play essential roles as signalling molecules for the immune system, such as the SCFAs and tryptophan metabolites. More specifically, a polysaccharide (PSA) produced by *Bac. fragilis* was shown to promote cellular and physical maturation of the developing immune system in mice [148]. Interestingly, this species is an early colonizer of the infant gut [149] and plays an important role in the development of the infant immune system. A 15 kDa protein produced by *F. prausnitzii*, a commensal microbe deficient in Crohn’s disease patients, was shown to have anti-inflammatory properties, decreasing activation of the NF-κB pathway and also prevented colitis in an animal model [150]. M-cells are specific phagocytic epithelial cells which sample particulate antigens [151]. Rios et al. [151] showed that efficient induction of IgA is brought about by M-cell sampling of commensal microbes. Paneth cells (a specialized intestinal epithelial lineage) sense enteric bacteria through activation of the MyD88-dependent toll-like receptor which results in the induction of several antimicrobial factors that are essential for controlling bacterial translocation across the intestinal barrier [152]. Colonisation by segmented filamentous bacteria was shown to induce the maturation of T-cell responses in a gnotobiotic mouse model, suggesting these microbes could play a role in the postnatal maturation of the gut immune system [153]. These studies provide a snapshot of how commensals modulate host immunity. The importance of the microbiota for host immunity can also be appreciated from the consequences of its absence in germ free animals. Indeed, germ free animals exhibit reduced expression of IgA and antimicrobial peptides and are deficient in Peyers’ patches [128,151,154,155]. 

Goblet cells are specialised epithelial cells that secrete mucus, composed primarily of O-glycosylated proteins called mucins, resulting in the formation of a mucus layer whose composition and density is influenced by the commensal microbiota [128,156,157]. The mucus layer creates a protective barrier for the epithelial cells making it difficult for pathogens to gain access to epithelial cell receptors [127]. Germ free mice have been shown to have an extremely thin colonic mucus layer which can be restored to levels observed in conventional mice following exposure to bacterial products, including peptidoglycan and lipopolysaccharide [158]. Mice treated with the antibiotic metronidazole were shown to have a thinning of the mucus layer which correlated with an increased attachment of the mouse pathogen *Citrobacter rodentium* [159]. Certain commensals have been shown to modulate mucin gene expression and glycosylation patterns [160,161,162]. This may be achieved through the activity of SCFAs which have been shown to increase expression of mucin-associated genes [163]. Furthermore, the SCFA butyrate provides energy for the epithelial cells and has also been implicated in enhancing the intestinal barrier by up-regulating the tight junction protein, Claudin-1 [87].

## 4. Low Diversity of the Microbiota Caused by Poor Quality Diet Linked With Risk of Infections and Inflammation Predominantly

Dysbiosis is a term used to describe imbalances in gut microbiota communities and is linked with disease when these imbalances negatively impact microbiota functions required for health or when they promote disease occurrence [164]. For a number of these diseases, the dysbiotic state is manifested as a reduction in microbial diversity and often an increase in facultative anaerobes relative to the ‘healthy’ gut microbiota [164], such as IBD [165,166,167], cancer [168], liver disease [169,170] and recurrent *C. difficile* infection (CDI) [171]. Such diseases tend to primarily inflict those living a Westernised lifestyle and consuming a Western diet which is characterised by low fruit and vegetable intake and high consumption of animal-derived protein (meat and processed meat), saturated fats, refined grains, sugar, salt, alcohol and corn-derived fructose [172,173,174,175] (Figure 1). Whether these changes in microbial diversity are the cause or consequence of these diseases is as yet unconfirmed, although having reviewed a number of studies, Mosca et al. [176] suggest the argument for a causal effect is strong in the case of several human conditions, a stance we concur with in terms of inflammation and infection based on the body of research presented here.

That diet influences the composition of the gut microbiota has been confirmed in several studies [177,178,179,180,181,182,183]. Even short-term dietary changes (four days) have been shown to alter human gut microbiota composition [178]. In this particular study, David et al. [178] reported that the entirely animal-based diet (meats, eggs and cheeses) had a greater impact on gut microbiota composition than the plant-based diet (grains, legumes, fruits and vegetables), resulting in a decrease in plant polysaccharide-metabolising Firmicutes (*Roseburia, Eubacterium rectale*, and *Ruminococcus bromii*) and an increase in the abundance of bile-tolerant microorganisms presumably owing to the increase in bile acid secretion as a result of high fat intake [184]. Indeed, the animal-based diet significantly increased the levels of faecal deoxycholic acid (DCA), a secondary bile acid produced by microbial dehydroxylation of bile, and has been shown to promote liver cancer in mice [185], and inhibit the growth of Bacteroidetes and Firmicutes members in rats [186]. Microbial genes required for DCA production exhibited significantly higher expression on the animal-based diet. Furthermore, the abundance and activity of the sulfite-reducing bacterium, *Bilophila wadsworthia*, was also shown to be increased on the animal-based diet.

In mouse models, this particular microorganism has been shown to cause IBD, which is thought to be due to its production of hydrogen sulfide which inflames intestinal tissue [187]. Overall microbial gene expression was strongly linked to diet, thus as expected, the animal-based diet resulted in lower levels of carbohydrate fermentation end-products, the SCFAs. Despite the increase in dietary fibre, four days of the plant-based diet did not increase the microbiota diversity of participants, most likely owing to the short time frame. Two days after the animal-based diet ended, the gut microbiota of participants reverted to their original structure. In a study investigating the associations between long-term dietary habits and lifestyle, and short-term dietary changes, with gut microbiota composition, Klimenko et al. [179] reported that alpha diversity was positively linked to the number of vegetables consumed in long-term dietary patterns. 

In certain rural regions of the world, communities continue to live a traditional lifestyle and thus consume diets resembling those of our early ancestors which are naturally high in fibre. For example, inhabitants of a rural African village in Burkina Faso still consume a high fibre diet similar to that of early human settlements at the time of the birth of agriculture. A comparative study examining the gut microbiota composition of the children of Burkina Faso versus European children (from Florence, Italy) consuming a Western diet (age of participants, 1−6 years) revealed a significantly higher richness and biodiversity in the gut microbiota of the Burkina Faso group [180]. The African children also displayed an enrichment of Bacteroidetes and depletion of Firmicutes relative to their European counterparts. Within the Bacteroidetes, the genera *Prevotella* and *Xylanibacter* were uniquely abundant in the African children being absent in the European children. These specific bacteria harbor genes for cellulose and xylan hydrolysis. In contrast, the potentially pathogenic *Enterobacteriaceae* (*Shigella* and *Escherichia*) were significantly over-represented in European children compared to their African counterparts. Furthermore, SCFAs were significantly more abundant in the African children. The authors hypothesise that the gut microbiota of the Burkina Faso children evolved with their polysaccharide-rich diet and protects them from inflammation and non-infectious colonic diseases. The traditionally-living Hadza people of Tanzania are one of the last hunter-gatherer communities in the world. A recent study reported no evidence of cardiovascular disease risk factors in this population [188] and older studies reported that this group of people had relatively low rates of metabolic diseases, infectious diseases or nutritional deficiencies compared to other settled groups in the surrounding regions [189,190,191]. A comparison of the gut microbiome of the Hadza people with an Italian urban cohort revealed higher levels of microbial richness and diversity in the Hadza group [192]. These studies suggest that the Western microbiota, even in a healthy person, may in fact be dysbiotic in terms of microbial diversity owing to the low consumption of MACs, and predisposes its host to a range of diseases, particularly those which are characterised by an inappropriate immune response, a theory which has been proposed by Sonnenburg and Sonnenburg [70]. 

The link between inflammation and low microbiota diversity was corroborated in a study involving 123 non-obese and 169 obese Danish individuals [193]. Within this collective group of 292 subjects, two groups could be differentiated by the number of gut microbial genes and thus bacterial richness. The ‘low gene count’ (LGC) group represented 23% of the total population studied and included a significantly higher proportion of obese subjects. The LGC group was characterised by a more pronounced inflammatory phenotype, marked overall adiposity, insulin resistance, and dyslipidaemia. Obese individuals within the LGC group were found to gain more weight over time. Only a few bacterial species were sufficient to distinguish between the LGC group and the ‘high gene count’ (HGC) group but interestingly anti-inflammatory species, such as *F. prausnitzii* [194] were more prevalent in HGC individuals while potentially pro-inflammatory species associated with IBD, *Bacteroides* and *Ruminococcus gnavus* [195,196,197], were more frequently found in LGC individuals. In an accompanying intervention study involving 49 obese and overweight subjects of whom 40% was defined as LGC, a similar phenomenon in terms of clinical parameters and inflammatory status was observed such that the authors concluded that LGC individuals are at increased risk of obesity-associated co-morbidities [177]. Members of the LGC group were found to consume fewer fruits and vegetables and fewer fishery products than the HGC group. An energy-restricted diet with increased fibre intake for six weeks resulted in an increase in microbial gene richness in the LGC group which approached but remained significantly different to that of the HGC group. Both groups showed a loss in body fat mass and an improvement in clinical phenotypes (lipid and insulin levels and insulin resistance) and a trend towards a decrease in inflammation (as measured by highly sensitive C-reactive protein) though the effects were more pronounced for the HGC group. Although this was a short-term intervention study, it suggests that measures of gene richness and microbial diversity may help predict the efficacy of interventions. Furthermore, it seems that long term improvements to dietary habits may be required to improve and stabilise gut microbial diversity which agrees with the observations of Klimenko et al. [179] who reported considerable correlations between long-term dietary habits and gut community structure. 

Loss of microbial diversity is also associated with increased risk of infection, presumably due to loss of colonisation resistance. For example, human studies have shown that the presence of the gut pathogen *C. difficile* is associated with decreased gut microbiota diversity in CDI patients [198,199,200,201], as well as in asymptomatic carriers [199]. Gu et al. [201] also reported a dramatic increase in endotoxin-producing opportunistic pathogens and lactate-producing phylotypes in CDI patients. Community richness and diversity were significantly lower in the gut microbiota of methicillin-resistant *Staphylococcus aureus* (MRSA)-positive patients compared to individuals without MRSA [202]. The alpha diversity of the gut microbiota of children suffering from acute infectious diarrhoea caused by rotavirus was significantly less diverse than those of healthy children [203]. In this case, probiotic intervention for five days resulted in recovery from diarrhoea. By day 3, diarrhoea symptoms had ceased, by days 10 and 30 after the intervention, microbiota diversity had increased to the point that it was no longer significantly different from healthy children. Future studies are required to determine the exact role of the microbiota in diarrhoea-related processes. Other viral infections have also been associated with low-diversity dysbiosis, including hepatitis C [204] and HIV [205]. 

While the exact mechanisms underlying the link between low microbial diversity, diet and disease are not fully understood, SCFAs undoubtedly play a role given that low MAC diets are directly linked to low SCFA levels [70,180,182]. Livanos et al. [206] reported a significant decrease in the proportion of the SCFA-producing Clostridial Clusters IV/XIVa in intensive care unit patients 72 h following hospital admission which was associated with reduced gut microbiota diversity and community stability over time. Simultaneously, the facultative anaerobe *Enterococcus* significantly expanded. These changes were associated with receipt of broad-spectrum antibiotics. The depletion of SCFAs and in particular butyrate producers has been reported in cases of CDI and asymptomatic carriage of *C. difficile*, in sufferers of nosocomial diarrhoea and in MRSA-positive patients [199,200,201,202]. The potential importance of butyrate in maintaining an oxygen-depleted environment in the lumen and impeding colonisation by facultative anaerobes has already been mentioned [133]; however, its specific role, if any, in colonisation resistance against *C. difficile* infection has not yet been elucidated and thus the viable bacteria themselves are most likely the responsible agents [200,207]. Indeed, the loss of a specific butyrate-producing species, *C. scindens*, a member of the *Clostridium* XIVa clade [208], was directly associated with susceptibility to *C. difficile* infection in a mouse model as a result of antibiotic treatment [144]. Administration of *C. scindens* alone or in combination with three other bacteria to antibiotic-treated mice ameliorated CDI. In this case, secondary bile acids produced by *C. scindens* were found to be responsible for the anti-*C. difficile* effect. However, a recent study reported the presence of *C. scindens* and *C. difficile* in the same stool sample and suggested that the former does not inhibit the latter [209] but the study does not provide data on bile acid profiles or the 7-α -dehydroxylating activity of *C. scindens*, the enzyme responsible for secondary bile acid production. 

Furthermore, Sonnenburg et al. [210] showed that in the absence of dietary polysaccharides, a human gut microbe turned to host mucus glycans as a nutrient source. Mouse models have shown that a defective mucus barrier enables contact between epithelial cells and bacteria resulting in spontaneous colitis in mice [211], a feature shared with sufferers of ulcerative colitis [212]. But a direct connection between dietary fibre and the status of the colonic mucosal barrier was more recently presented in a mouse study by Desai et al. [213]. In this study, a dietary-fibre deprived gut microbiota resorted to host-secreted mucus glycoproteins, which resulted in the erosion of the colonic mucus barrier and enabled the gut pathogen *Cit. rodentium* greater access to the epithelial cells, resulting in lethal colitis. In another study, mice fed a Western style diet presented with an altered gut microbiota composition that resulted in increased permeability and reduced growth rate of the inner mucus layer compared with mice fed a CHOW diet [214]. However, administration of *Bif. longum* or the fibre inulin prevented mucus defects. Inulin prevented penetrability of the inner colonic mucus layer while *Bif. longum* restored mucus growth. 

The Winning the War on Antibiotic Resistance (WARRIOR) project which is being undertaken by researchers from the University of Wisconsin aims to investigate the relationship between dietary fibre intake, the gut microbiota and colonisation by multi-drug resistant microorganisms using 600 randomly selected Wisconsin residents over the age of 18 of which the main results will be published in a peer-reviewed journal [215]. The results of this study should help us to further delineate the role of diet and the microbiota in protecting against pathogen invasion.

## 5. Diet-Derived Microbial Metabolites—Many Are Beneficial but Specific Metabolites Are Associated with Risk of Metabolic Disease

Metabolic disease refers to any disease in which the normal metabolic processes in the body are disrupted and examples include obesity, type 2 diabetes and metabolic syndrome, all of which are risk factors for cardiovascular disease. Obesity and overweight are described as abnormal or excessive fat accumulation that negatively impacts health [216] and according to the WHO, obesity has tripled since 1975. Indeed, in 2016, 39% of adults aged 18 years and over were described as overweight and 13% as obese [216]. A recent study found that body mass index (BMI) had a J-shaped association with overall mortality among 3.6 million adults in the UK [217]. Obesity results from ingestion of excess energy which does not get expended and is a strong risk factor for type 2 diabetes, the latter of which is characterised by high blood sugar levels, insulin resistance and relative lack of insulin [218]. In 2014, the WHO [219] estimated that 422 million adults had diabetes, of which the majority were inflicted with type 2 diabetes. Diabetic dyslipidaemia which describes high levels of triglycerides in blood plasma, increased levels of small, dense, low-density lipoprotein (LDL) cholesterol and decreased levels of high-density lipoprotein (HDL) cholesterol is associated with insulin resistance and is a risk factor for cardiovascular disease in diabetic individuals [218,220]. Metabolic syndrome describes the abnormal metabolism of glucose and lipids and is characterised by abdominal obesity along with two or more of the following factors: reduced HDL cholesterol, elevated triglycerides, high blood pressure and increased fasting blood glucose [218]. Metabolic syndrome is also a risk factor for type 2 diabetes along with cardiovascular disease [221].

The association between the gut microbiota and regulation of energy storage in the body was initially documented in mice. Indeed, germ free mice are protected against the obesity that usually develops after consuming a Western-style, high fat, sugar rich diet [222]. Conventionalisation of adult germ-free mice with a normal microbiota from conventionally raised animals resulted in a 60% increase in body fat and insulin resistance within 14 days [223]. The obesity-resistance phenotype of germ-free animals fed a high fat diet was attributed to increased excretion of lipids and reduced calorie consumption [224]. The interactions between high fat diet and the gut microbiota in mice were shown to promote pro-inflammatory changes in the small intestine which preceded weight gain and obesity and also showed significant associations with obesity progression and development of insulin resistance [225]. In terms of microbiota composition, an increased abundance of Firmicutes and decreased abundance of Bacteroidetes has been reported in obese mice compared to their lean counterparts [226]. It has been suggested that such a microbiota may be more efficient at generating energy in the form of SCFAs from the diet thus contributing to weight gain [227]. Differences in gut microbiota composition and functionality have also been recorded in humans between obese and lean people. Ley et al. [228] reported a decrease in the relative proportion of Bacteriodetes in obese people relative to their lean counterparts and which was found to increase on calorie-restricted diets. This observation was also corroborated by Turnbaugh et al. [229] who also noted a higher proportion of Actinobacteria in obese subjects. Furthermore, faecal SCFA concentrations tend to be higher in obese individuals [230,231,232,233]. The obese gut microbiota has also been characterised by reduced diversity and altered representation of bacterial genes and metabolic pathways [229]. More recently, a systematic review investigated the link between the gut microbiota and low-grade inflammation, a hallmark of obesity which plays a major role in atherosclerotic cardiovascular disease in humans [234]. Fourteen, mostly observational studies were included with *n* = 10 to 471 participants. Higher white blood cell counts and high sensitivity C reactive protein levels were associated with lower gut microbial diversity, and the abundances of *Bifidobacterium, Faecalibacterium, Ruminococcus* and *Prevotella* were inversely linked to different markers of low-grade inflammation. Compositional and functional differences have been observed in the gut microbiota of type 2 diabetes individuals which included a moderate degree of microbial dysbiosis, reduction in the abundance of some universal butyrate-producing bacteria, an increase in opportunistic pathogens, and an increase in microbial functions conferring oxidative stress resistance [235,236]. In apparently healthy participants, Kashtanova et al. [237] reported an association between compromised metabolic status, lower gut microbial alpha diversity, and higher representation of opportunistic pathogens and as a consequence propose that the “gut-heart axis” may be involved in the very early stages of cardiovascular disease. These studies clearly illustrate the link between the gut microbiota and metabolic disease of which certain microbial metabolites undoubtedly play a significant role [238]. 

The beneficial role of microbially-produced SCFAs in energy regulation of the host, insulin sensitivity and glucose and lipid metabolism has been presented in numerous studies which have been the topic of recent reviews [76,238,239]. However, certain microbial metabolites have been shown to be detrimental to health and may be significant contributors to the development and progression of metabolic diseases in the host [97]. Dietary-derived microbial metabolites are of particular interest given that dietary modulation and probiotic and prebiotic inventions offer viable routes to alter microbial metabolite signatures. 

Trimethylamine (TMA) is a microbial metabolite generated from dietary choline, as well as the choline derivative betaine, lecithin, and L-carnitine [238]. Red meat, dairy products, eggs and salt-water fish are rich sources of choline, lecithin and carnitine [240]. Gut microbes known to be involved in TMA production include members of the families *Deferribacteraceae, Anaeroplasmataceae, Prevotellaceae*, and *Enterobacteraceae* [240,241,242,243]. The microbially-produced TMA crosses the intestinal epithelial cells, enters the circulation and is transported to the liver where it is converted to the uremic toxin, trimethylamine-*N*-oxide (TMAO) mainly by the enzyme flavin-containing monooxygenase-3 (FMO3) [244]. Elevated plasma TMAO levels derived from microbial metabolism of the lipid phosphatidylcholine have been identified as a risk factor for cardiovascular disease in humans [245]. In the same study, dietary supplementation of mice with TMAO or choline was found to promote atherosclerosis and studies with germ free mice indicated the significance of the gut microbiota and dietary choline in TMAO production, increased macrophage cholesterol accumulation and foam cell formation. Gut microbial-derived production of TMAO from L-carnitine, which is abundant in red meats, has also been confirmed in humans, suggesting a mechanism for the link between dietary red meat ingestion and accelerated atherosclerosis [241]. Interestingly, in the same study dietary supplementation of mice having intact intestinal microbiota with TMAO was shown to reduce the expression of key liver enzymes involved in bile acid synthesis, as well as a number of liver bile acid transporters which resulted in significant decreases in the total bile acid pool and decreased reverse cholesterol transport. The interaction between consumption of dietary components, TMAO levels and various disease endpoints may be affected more by interindividual variations in TMAO production than consumption of the dietary components themselves [246]. Additionally, there is significant variability in one of the key enzymes (FMO3) involved in TMAO production (can be 20–30× difference between individuals) making it difficult to generalise and draw conclusions about diet and disease endpoints in regard to TMAO production. TMAO has also been shown to induce inflammatory markers in mice and in human aortic endothelial cells and vascular smooth muscle cells owing to activation of the mitogen-activated protein kinase, extracellular signal kinase, and the NF-κB signalling cascade, and it promoted recruitment of activated leukocytes to endothelial cells [247]. It has also been shown to activate the NLRP3 inflammasome [248]. The link between plasma TMAO levels and the gut microbiota in humans was further confirmed following phosphatidylcholine challenge (ingestion of two hard-boiled eggs and deuterium [d9]-labelled phosphatidylcholine) in the presence and absence of broad-spectrum antibiotics [249] whereby plasma TMAO levels were suppressed after antibiotic administration but reappeared after withdrawal of antibiotics. Furthermore, the study confirmed the link between elevated TMAO plasma levels and increased risk of incident major adverse cardiovascular events. The metabolite has also been associated with disease severity and survival of patients with chronic heart failure [250], and was shown to cause platelet aggregation in humans receiving a choline supplement for two months thus revealing a link between TMAO and risk of thrombosis [251]. A systematic review and meta-analysis of 19 prospective studies found an association between elevated plasma TMAO levels and its precursors (L-carnitine, choline or betaine) with increased risks of major adverse cardiovascular events and death, independent of traditional risk factors [252]. Higher plasma TMAO levels were also found to be associated with diabetes [253]. Interestingly, however, a structural analogue of choline, namely 3,3-dimethyl-1-butanol (DMB), was able to non-lethally inhibit TMA formation by cultured microbes, by physiological polymicrobial cultures (intestinal contents, human faeces), and reduce TMAO levels in mice fed a high carnitine or choline diet [254]. In the same study, DMB also inhibited choline diet-enhanced endogenous macrophage foam cell formation and development of atherosclerotic lesions in apolipoprotein e-/- mice without altering circulating cholesterol levels. This study provides a specific example of a therapeutic approach towards the treatment and prevention of a metabolic disease through direct manipulation of gut microbiota metabolism. 

The aromatic amino acids tyrosine and phenylalanine are metabolised by intestinal bacteria to the phenolic compound, *p*-cresol, which is a uremic toxin [255,256]. This compound is detoxified in the liver and colon where it is sulfated to become *p*-cresol-sulfate [257,258]. The absence of *p*-cresol-sulfate in the plasma of germ-free mice provides further evidence of the role of the gut microbiota in its production [116]. Under normal conditions *p*-cresol-sulfate enters the circulation and ends up in the urine where it is excreted from the body [256]. Information regarding the specific intestinal microbes responsible for *p*-cresol generation is still unclear, but *in vitro* studies have shown that members of the following families can produce it; *Clostridiaceae, Eubacteriaceae, Lachnospiraceae, Ruminococcaceae, Staphylococcaceae, Veillonaceae, Bacteroidaceae, Porphyromonadaceae, Bifidobacteriaceae,* and *Fusobacteriaceae* [256]. However, in patients suffering from chronic kidney disease, excretion is impaired. Chronic kidney disease and cardiovascular disease are closely associated and risk of death from cardiovascular disease is much higher in patients with end-stage renal disease [259] and evidence strongly indicates that *p*-cresol-sulfate is a causal link. *p*-Cresol-sulfate has been shown to disturb the normal cell cycle in a mouse preadipocyte cell line, to induce apoptosis, inhibit differentiation of preadipocyes to mature adipocytes, and to decrease glucose uptake at baseline and after insulin stimulation [260]. It has been shown to have a pro-inflammatory effect on unstimulated leucocytes [261]. It induced shedding of endothelial microparticles in human endothelial cells, a marker of endothelial damage [262]. It induced oxidative stress in both endothelial and vascular smooth muscle cells at a concentration representative of that found in dialysis patients, and it increased contraction of mouse thoracic aorta and eventually led to inward eutrophic remodelling, a feature of uremic vasculopathy, suggesting it may contribute to hypertension and cardiovascular mortality in chronic kidney disease [263]. In nephrectomised mice, it promoted cardiac apoptosis which was at least partly attributed to the induction of NADPH oxidase activity and production of reactive oxygen species [264]. Poesen et al. [265] showed that intestinal uptake of *p*-cresol in patients with chronic kidney disease associated with cardiovascular disease independent of renal function. 

Indoxyl sulfate is another uremic toxin derived from gut microbial metabolism of the essential amino acid, tryptophan [266]. In this case, bacterial trytophanases generate indole from tryptophan which is transported to the liver where it is hydroxylated and O-sulfated to produce the uremic toxin. Like *p*-cresol, serum indoxyl sulfate is problematic for chronic kidney disease patients. Barreto et al. [267] reported that baseline indoxyl sulfate levels presented an inverse relationship with renal function and a direct relationship with aortic calcification in chronic kidney disease patients. While the causative relationship between indoxyl sulfate and the pathology of cardiovascular disease has not yet been proven, studies to date suggest its actions are linked to multiple NADPH oxidase-mediated redox signalling pathways which have been implicated in different forms of cardiovascular disease pathophysiologies, including coronary calcification, atherosclerotic vascular disease, arrhythmia, and chronic heart failure [268]. Similarly, the microbial metabolite phenylacetylglutamine, which is derived from the microbial conversion of phenylalanine [269], has been shown to be a strong and independent risk factor for cardiovascular disease and mortality in chronic kidney disease patients [270]. 

The branched chain amino acids (BCAAs), leucine, isoleucine and valine represent three of the nine essential amino acids. As well as being building blocks of protein synthesis, the BCAAs are involved in nutrition metabolism, regulation of energy homeostasis, gut health, immunity and disease [271]. Our gut microbiome has been shown to be enriched with genes involved in the biosynthesis of essential amino acids, including the BCAAs [272]. While BCAAs are not precisely ‘diet-derived’ microbial metabolites, a recent study linking microbial BCAAs to glucose intolerance and insulin resistance renders them worthy of mention. Indeed, Pedersen et al. [273] reported that the serum metabolome of insulin-resistant individuals was characterised by increased levels of BCAAs which correlated with a gut microbiome with enriched biosynthetic potential for BCAAs. Transfer of *Prevotella copri* to mice, one of the main species driving the association between BCAA synthesis and insulin resistance, increased circulating levels of BCAAs, induced insulin resistance and aggravated glucose intolerance. The authors state that further studies are required to determine how the intestinal BCAAs enter the bloodstream and from what intestinal location. However, elevated circulating BCAA levels and their breakdown products have been positively associated with insulin resistance in humans. A metabolic signature related to BCAA catabolism in obese subjects which included circulating BCAAs and downstream products of their catabolism (acylcarnitines and glutamate) was identified in “healthy” obese subjects free of diabetes or other serious illnesses [274]. These obese subjects were also more insulin resistant compared to their lean counterparts. In the same study, animals fed a high fat diet supplemented with BCAAs had reduced food intake and weight gain compared to the high fat diet alone but became equally insulin resistant as those on the high fat diet. The authors proposed a pathway of dysregulated BCAA metabolism as a result of overnutrition of protein which leads to insulin resistance, glucose intolerance and eventually type 2 diabetes. A systematic review of 23 studies involving 20,091 participants found that circulating BCAAs are a useful biomarker for detection of insulin resistance and diabetic risk later on [275]. A causal role of BCAA metabolism in the aetiology of type 2 diabetes was reported following a large-scale human genetic and metabolomic study [276]. Therefore, modulation of the microbiota to influence BCAA levels could represent a potential route towards the prevention of insulin resistance and its associated aetiologies. 

As we have seen from these studies, metabolomics alongside metagenomics is a powerful tool towards identifying diet-derived microbial metabolites involved in human disease. These metabolites not only serve as biomarkers of disease occurrence and disease risk but may provide new routes towards disease management through modulation of the intestinal microbiota. 

## 6. Role of Diet Type in Shaping the Microbiome

The Mediterranean diet is characterised by high intake of non-refined grains, legumes, and a large diversity of fresh vegetables and fruits daily; yogurt and milk are consumed only a few times per week; there is reduced intake of fish and seafood, eggs, white meat and high fat dairy products (a few times per week); red meat is consumed only a few times per month; alcohol intake is also minimised; dietary lipids are mainly derived from olive oil [277]. Epidemiological studies and randomized clinical trials have shown that consumption of a Mediterranean diet bestows protective effects against several diseases, including obesity, diabetes, hypertension, cardiovascular disease, stroke, numerous cancers, allergic diseases, as well as Parkinson’s and Alzheimer’s disease (recently reviewed by Tosti et al. [278]). Based on evidence to date, Tosti et al. [278] concluded that the health benefits of the Mediterranean diet are due to its lipid-lowering effects, protection provided against inflammation, oxidative stress and platelet aggregation, modulation of growth factors and hormones involved in cancer pathogenesis, inhibition of nutrient sensing pathways by specific amino acid restriction and gut microbiota metabolites which influence host metabolic health. Indeed, the accumulating evidence suggests that the Mediterranean diet modulates gut microbiota composition and functionality resulting in microbiome and metabolome which differs from that of the Western diet and reduces risk of disease. For example, consumption of the Mediterranean diet for one year by obese men (*n* = 20) significantly decreased the genera *Prevotella* and increased *Roseburia* and *Oscillospira* [279]. *Roseburia* is a known butyrate-producer with immune maintenance and anti-inflammatory properties [280] and *Oscillospira* is positively associated with leanness and health [281]. Long-term consumption of the diet also increased the abundance of *Parabacteroides distasonis* [279], a strain recently shown to block colon tumor formation in high fat diet-fed azoxymethane-treated mice [282]. Faecal metabolite changes linked to amino acid, peptide and sphingolipid metabolism were reported on the Mediterranean diet and an improvement in insulin sensitivity was noted [279]. The same group investigated the effect of chronic consumption of the Mediterranean diet for 2 years in obese patients with severe metabolic disease (*n* = 33), obese patients without metabolic dysfunction (*n* = 32) and non-obese subjects (*n* = 41) [283]. Obese patients with severe metabolic disease showed a marked dysbiosis in their gut microbiota which was reversed following consumption of the Mediterranean diet; furthermore, there was a significant reduction in plasma triglyceride levels and a trend for glucose depletion in the same group. In this case, consumption of the Mediterranean diet for two years increased the abundance of *Bacteroides* and *Prevotella* which make up the Bacteriodetes phylum, which has previously been reported to be reduced in obese people [228]. In addition, genera with saccharolytic activity, including *Faecalibacterium*, *Roseburia* and *Ruminococcus* were also increased. Interestingly, such changes were not observed for non-obese individuals or obese individuals without metabolic syndrome. De Filippis et al. [284] reported that adherence to the Mediterranean diet in a cohort of Italian individuals (*n* = 153) was associated with increased levels of faecal SCFAs which correlated with enrichment of members of Firmicutes and Bacteroidetes. In contrast, low adherence to the diet (particularly observed in omnivores) was associated with increased levels of urinary TMAO which correlated with L-*Ruminococcus* (*Ruminococcus* genus assigned to *Lachnospiraceae* family). Interestingly, the mucin-degrading bacterium *Ruminococcus torques* has been shown to be increased in transgenic mice representative of Crohn’s disease and receiving a high fat/high sugar Western diet [285]. These mice had decreased mucus layer thickness and increased intestinal permeability. Higher adherence to the Mediterranean diet in a Spanish cohort (*n* = 74) was also associated with higher levels of *Clostridium* cluster XVIa and *F. prausnitzii* [286]. *Clostridium* cluster XVIa contains the *Blautia coccoides* group which are known butyrate producers [287], are involved in secondary bile acid production [288], as well as in the formation of T-regulatory cells [289]. *F. prausnitzii* is now recognised as one of the most abundant butyrate producers in the human gut microbiota and has anti-inflammatory properties [194]. Interestingly, Mitsou et al. [290] reported that fast food consumption in an adult population resulted in a microbiota with suppressed representation of *Lactobacillus* and butyrate-producing bacteria while adherence to the Mediterranean diet led to lower faecal *E. coli* counts, a higher *Bifidobacterium*: *E. coli* ratio and greater molar ratio of acetate (*n* = 120). The study also reported increased levels and prevalence of *Candida albicans* due to the Mediterranean diet which the authors suggest could be linked to the ingestion of foodstuffs that are carriers of yeast. Overall, the studies presented here strongly indicate that increased SCFA production as a result of gut microbiota modulation on the Mediterranean diet plays a key role in its beneficial health effects. 

Another “trendy” diet type that shapes the microbiome is the gluten-free diet. De Palma et al. [291] showed that a 30-day gluten-free diet decreased populations of *Bifidobacterium* and *Lactobacillus* while increasing unhealthy bacteria, such as *E. coli* and *Enterobacetriaceae* that could further increase the risk of infection by opportunistic bacteria. Others, such as Bonder et al. [292] found reductions in *R. bromii* and *Roseburia faecis* coupled to increased *Victivallaceae* and *Clostridiaceae* in individuals consuming a gluten-free diet. Whilst a gluten-free diet is an effective therapy for treating patients diagnosed with celiac disease it is often associated with a number of health issues and nutritional deficiencies. Interestingly, over the last 5–10 years, there has been an increasing number of healthy, non-celiac individuals following gluten-free diets and do so under the impression that such a diet is healthier for them when paradoxically it may be increasing their risk of the similar health issues and nutrient deficiencies as those commonly seen in those with celiac disease.

## 7. Dietary Components (Protein, Carbohydrate, Fat) Influencing Microbiota Composition

Protein, carbohydrate and fat are macronutrients required in large amounts to maintain bodily functions and to provide energy for the body. FAO/WHO recommends that daily dietary fat should not exceed 30% of total energy intake, protein should account for 10–15%, and carbohydrate should account for the remainder (between 55–75%). Dietary fibre intake is recommended at 25 g/day for women and 38 g per day for men [293]. However, the proportions of these macronutrients in the typical Western diet are skewed towards high-energy and low nutrient-dense foods containing large amounts of saturated fat and sugar, and fibre intake has been reported to be as low as 15 g/day in 90% of the population in developed countries [294]. Furthermore, each of these macronutrients contains different types and studies have shown that these differences along with quantity consumed can dramatically influence gut microbiota composition and functionality. 

### 7.1. Fat

Dietary fat is composed mainly of triglycerides where each triglyceride molecule contains a glycerol backbone with three fatty acids attached [295]. Some dietary fats reach the colon after escaping absorption in the small intestine. Indeed, we have already seen that an animal-based diet high in fat and protein alters the gut microbiota resulting in an increase in the abundance of bile-tolerant microorganisms, such as *Bilophila*, *Alistipes* and *Bacteroides* [178] and long-term consumption selects for a *Bacteroides* enterotype [33]. Recently, Agans et al. [296] directly investigated if typically consumed dietary fatty acids alone could sustain the growth of the human gut microbiota using an *in vitro* multi-vessel simulator system of the human small intestine. Switching from a balanced Western diet type medium to one lacking carbohydrates and proteins resulted in substantial changes to the microbiota and metabolites produced. Several specific genera increased in abundance, including *Alistipes, Bilophila* and several genera within the class Gammaproteobacteria. Increased abundance of *Alistipes* has been correlated with a greater frequency of pain in paediatric patients with irritable bowel syndrome [297], has been found in patients with major depressive disorder [298] and has been associated with type 2 diabetes [236]. We have already seen that *Bil. wadsworthia* was increased in humans consuming the animal-based diet [178] and is associated with inflammation owing to its hydrogen sulfide-producing capabilities [187]. *E. coli* is a member of Gammaproteobacteria which also harbours many pathogenic members, including *Salmonella*, *Yersinia*, *Vibrio*, and *Pseudomonas* [299]. The abundances of well known glycan and protein degraders, including *Bacteroides, Clostridium* and *Roseburia* decreased in the fats-only medium and there was also a decrease in the production of antioxidants and SCFAs, a concomitant decrease in the abundance of glycan-degradation genes, and an increase in the abundance of genes coding for fatty acid degradation enzymes and anaerobic respiratory reductases [296]. Overall, the results show that the gut microbiota is capable of utilising dietary fats typical of the Western diet but the resulting changes may negatively impact human health. In mice, Murphy et al. [300] showed that high fat feeding of mice was more influential for altering gut microbiota composition than genetically induced obesity, generating a progressive increase in Firmicutes which reached statistical significance. In another study, high fat feeding of mice was shown to decrease the proportion of *Ruminococcaceae*, which are known to utilise plant polysaccharides, and to increase the proportion of *Rikenellaceae* [301], of which the genus *Alistipes* is a member. The results also suggested that high fat feeding altered overall cellular composition within the abundant bacterial groups, namely, *Bacteroidales* and *Lachnospiraceae* based on cecal chemical fingerprints analysis. Metaproteome analysis revealed a decrease in proteins involved in carbohydrate metabolism and a shift towards amino acid and simple sugar metabolism. This is in agreement with Yatsunenko et al. [24] who reported that metagenomes associated with the Western diet are enriched in amino acid and simple sugar degrading enzymes relative to African populations consuming diets rich in complex carbohydrates. More recently, Vaughn et al. [302] reported that a high fat diet fed to Sprague-Dawley rats increased the Firmicutes: Bacteriodetes ratio and increased proliferation of pro-inflammatory *Proteobacteria* which were found to be toxic to vagal afferent neurons in culture. Overall, the results suggest that high fat-diet-induced shifts in gut microbiota could disrupt vagal gut-brain communication ultimately leading to increased accumulation of body fat. 

However, dietary fat is not a homogenous macronutrient, since the structure and function of different fatty acids can vary greatly depending on chain length (6 to 24 carbons) and absence or presence of carbon-carbon double bonds [295]. Saturated fatty acids (SFAs) have no double bond, monounsaturated fatty acids (MUFAs) contain a single double bond, while polyunsaturated fatty acids (PUFAs) have two or more double bonds. Furthermore, the double bond can be in a *cis* or *trans* configuration depending on whether the hydrogens attached to the carbons in the double bond are on the same side or opposing sides of the molecule, respectively. Trans fats are produced commercially by the partial hydrogenation of unsaturated fats or naturally via biohydrogenation in ruminant animals [303]. Studies have shown that different fatty acids exert differential effects on the gut microbiota. For example, a high fat diet rich in saturated fat in the form of palm oil (PUFA: SFA ratio = 0.4) induced higher body weight gain and liver triglyceride content in mice compared with diets rich in polyunsaturated fats (olive oil, PUFA: SFA ratio = 1.1, or safflower oil, PUFA: SFA ratio = 7.8) [304]. Interestingly, the overflow of dietary fat to the distal intestine was greater on the palm oil diet compared with the others. Indeed, the high-saturated fat, palm oil diet reduced microbial diversity and increased the Firmicutes: Bacteroidetes ratio which the authors suggest may be due to antimicrobial effects of the saturated fats. A diet high in SFAs derived from milk fat, but not PUFAs derived from safflower oil, promoted expansion of the low abundance sulfite-reducing *Bil. wadsworthia* in mice which was associated with a pro-inflammatory immune response and increased incidence of colitis in genetically susceptible mice, but not wild type mice [187]. Due to their hydrophobicity, milk fats promote taurine-conjugation of hepatic bile acids which increases the availability of organic sulfur for use by bacteria, such as *Bil. wadsworthia*. In the same study, both the SFA- and PUFA-rich diets reduced microbiota richness compared with a low-fat diet. In this case, the low-fat diet promoted Firmicutes while the two high fat diets resulted in higher abundances of Bacteroidetes and low levels of Firmicutes. Interestingly, mice fed high fat diets derived from milk fat, or lard (both high in SFAs), or safflower oil (high in PUFAs) revealed that all three high fat diets induced dramatic and specific changes to gut microbiota composition which were associated with different host adipose tissue inflammatory/lipogenic profiles [305]. Safflower oil is rich in omega-6 PUFAs which are associated with a pro-inflammatory status [306]. Indeed, the safflower oil-rich diet resulted in a greater localized, tissue-specific pro-inflammatory effect compared to the milk fat and lard-based diets [305]. However, in both milk fat- and safflower oil-fed mice, Tenericutes were decreased, a phylum which has been shown to be reduced under inflammatory conditions [307,308] and Proteobacteria was increased, of which *Bil. wadsworthia* is a member. Consumption of extra virgin olive oil (rich in MUFAs and phenolic compounds), refined olive oil (rich in MUFAs but low in phenolic compounds), or butter (rich in SFAs and cholesterol) by mice resulted in differential effects on gut microbiota composition [309]. The butter-induced changes resembled those reported for obese individuals while the changes induced by extra virgin olive oil differed most from butter. In a more recent study, the same group investigated the effect of extra virgin olive oil on mouse gut microbiota compared to butter using more in-depth analysis [310]. Highest levels of systolic blood pressure were recorded for mice fed butter which correlated positively with *Desulfovibrio* sequences in faeces, which were significantly higher in mice fed butter compared to olive oil. Interestingly, *Desulfovibrionaceae* are sulfate reducers, which the authors suggest could be sustained by butter sulfate sources, whereas the authors suggest that the lower levels in the olive oil diet may be due to the higher presence of polyphenols. Olive oil had the lowest plasmatic insulin levels which correlated inversely with *Desulfovibrio*, and the lowest plasmatic leptin values which correlated inversely with *Sutterellaceae, Marispirillum* and *Mucilaginibacter dageonensis* which were significantly higher for olive oil. Mice fed the standard diet had the lowest total cholesterol levels which correlated positively with *Fusicatenibacter* and *Prevotella*, the latter of which has been associated with an improvement in glucose metabolism [311] but its relationship with cholesterol is less clear [310]. Interestingly, butter was also associated with an increase in *Alistipes indictintus* and *Pontibacter lucknowensis* which correlated positively with total cholesterol, ghrelin, insulin, body weight and HDL/LDL. Lam et al. [312] reported a rapid increase of hydrogen sulfide-producing bacteria in mice consuming a diet high in saturated fat which was not observed for mice fed high fat diets containing omega-6 or omega-3 PUFAs. In fact, these bacteria remained relatively stable in mice fed omega-6 PUFAs and were mostly reduced in mice fed omega-3 PUFAs. Likewise, Shen et al. [313] reported that mice fed a diet high in saturated fat had a greater abundance of three types of sulfidogenic bacteria (*Bil. wadsworthia, Desulfobulbus* and *Desulfovibrio*) primarily in the colonic mucosa compared to mice fed a low-fat diet after chronic feeding (20 weeks). In addition, high fat feeding increased intestinal inflammation by week 20, which was not observed at week 6. However, by week 6, high fat feeding had impaired the localization of the tight junction protein zonula occludens 1 at the apical area of the ileal epithelium which was also observed at week 20. The authors concluded that chronic high saturated fat feeding may contribute to chronic intestinal inflammation owing to microbial metabolic pathways. In humans at risk of metabolic syndrome, a high saturated fat diet, a high MUFA diet combined with high glycemic index (GI) carbohydrates, or a high MUFA diet combined with low GI carbohydrates caused a decrease in total bacteria and the high saturated fat diet increased faecal SCFA concentrations which the authors suggest could be due to lower absorption in the gut [314]. 

Interestingly, transgenic mice constitutively producing omega 3-PUFAs fed a high fat/high sucrose diet displayed higher phylogenetic diversity in the cecum compared to wild-type mice fed the same diet [315]. The transgenic mice were protected from obesity, glucose intolerance and hepatic steatosis and maintained a normal gut barrier function unlike wild-type mice. Transplantation of the faecal microbiota from the transgenic mice to the wild-type mice reversed their weight gain and normalised their glucose tolerance and intestinal permeability. The authors concluded that the omega-3 mediated changes to the gut microbiota were involved in the prevention of metabolic syndrome in the transgenic mice. 

Trans fats contain unsaturated fatty acids with at least one double bond in the *trans* configuration. In industry, they are formed during the partial hydrogenation of unsaturated fats (vegetable oils), a process performed to generate semi-solid fats for use in commercial cooking, margarines and manufacturing processes [316]. These fats, also referred to as partially hydrogenated oils (PHOs) have a long shelf life and can be customised to enhance palatability of sweets and baked goods and thus are an attractive ingredient for food manufacturers [316]. However, their consumption is associated with increased risk of disease, such as cardiovascular disease [303], type 2 diabetes [317], and Alzheimer’s disease [318]. Carvalho et al. [319] investigated the impact of PHOs on gut microbiota in mice in the presence of whey protein hydrolysate or casein as a protein source. The PHOs had minimal impact on the gut microbiota and were hardly able to invert the Bacteroidetes: Firmicutes ratio but preserved the normal pattern. 

Conjugated linoleic acid (CLA) is a collective term describing isomers of linoleic acid (LA) and while the latter has double bonds in the *cis* configuration located at carbons 9 and 12, CLA can have either the *cis* or *trans* configuration or both located along the carbon chain [320]. Unlike the *tran*s fats described earlier, CLA has been associated with numerous beneficial properties based primarily on results from animal models and cell lines, including protection against cancer, obesity and atherosclerosis, as well as immunomodulation which has been reviewed by Yang et al. [321]. The most bioactive isomers of CLA are *cis*-9, *trans*-11 CLA and *trans*-10, *cis*-12 CLA [321]. Supplementing mice with dietary *trans*-10, *cis*-12 CLA for eight weeks resulted in significantly lower proportions of Firmicutes and higher proportions of Bacteroidetes compared with the control group which received no supplementation [322]. Gut microbiota composition was significantly altered with higher proportions of the family *Porphyromonadaceae* and decreased abundance of families *Lachnospiraceae* and *Desulfovibrionaceae*. *Porphyromonadaceae* has been associated with non-alcoholic fatty liver disease (NAFLD) in mice [323], a manifestation of metabolic syndrome in the liver [324]. The family has also been associated with cognitive impairment in cirrhosis patients, whereas *Lachnospiraceae* abundance was reported to be lower in cirrhosis patients [325]. *Lachnospiraceae* has also been shown to protect mice against *C. difficile* colonisation [326] and most recently has been suggested to play a protective role against colorectal cancer [327]. *Desulfovibrionaceae* contains the sulfite-reducing species *Bil. Wadsworthia*. *Desulfovibrionaceae* has also been associated with impaired glucose tolerance and most serious metabolic syndrome phenotypes in mice [328]. Thus, the results of the study indicate that gut microbiota alterations following long-term supplementation of the *tr*ans-10, *cis*-12 CLA isomer could be detrimental to health. However, given that CLA supplements are commercially available, the authors suggest that fatty acid mixtures with equal proportions of CLA isomers or probiotics and prebiotics may balance such negative effects [322]. 

In a recent review investigating the available evidence on the impact of dietary fat on the gut microbiota and low-grade systemic inflammation and clinical implications for obesity, Cândido et al. [329] concluded that high fat diets and saturated fat should be avoided, while MUFAs and omega-3 PUFAs should be encouraged in order to regulate gut microbiota and inflammation towards promoting control of body weight/fat.

The ketogenic diet, defined as a high fat and low carbohydrate diet, has been effectively used as a therapeutic treatment for a number of neurological disorders, including epilepsy, Alzheimer’s disease, Parkinson’s disease, depression, autism, traumatic brain injury and depression, as examples [330]. Recently, researchers have revealed that the protective effects of the ketogenic diet could be mediated via the gut microbiota based on studies in mice [331,332]. The ketogenic diet administered by Ma et al. [331] consisted of 75.1% fat composed of SFAs, MUFAs and PUFAs. Following 16 weeks on the diet, mice revealed several neurovascular enhancements with potential to reduce the risk of Alzheimer’s disease which could be associated with the observed gut microbiota changes which included an increase in beneficial bacteria, including *Akkermansia muciniphila* and *Lactobacillus* and a reduction in pro-inflammatory microbes *Desulfovibrio* and *Turicibacter*. The gut microbiota changes observed by Olson et al. [332] in mice receiving the ketogenic diet were found to be required for the protective effects of the diet against acute electrically induced seizures and spontaneous tonic-clonic seizures. In this case, *Akkermansia* and *Parabacteriodes* were significantly increased and enrichment of, and gnotobiotic colonisation with these microorganisms were capable of restoring seizure protection in germ-free mice or mice treated with antibiotics. Additionally, in a mouse model of glioma, mice fed the ketogenic diet had slightly increased survival compared to mice fed the control diet and showed significant differences in several key microorganisms [333]. The ketogenic diet generally includes fats of all chain lengths [334] thus increases in blood ketone levels could be responsible for the observed gut microbiota changes [331]. 

### 7.2. Protein

It has been estimated that up to 25 g of protein, peptides and free amino acids enter the colon on a daily basis [335,336,337]. Colonic microbial digestion of these materials generates a range of end products, including SCFAs, indoles, amines, phenols, thiols, hydrogen sulfide, CO_2_ and H_2_, some of which are essential for health maintenance and some of which are detrimental [337]. Studies investigating the direct impact of protein on gut microbiota composition and functionality have shown that protein quantity, quality, processing history (which impacts on protein digestion, presentation and overall function) and source must be taken into consideration. For example, a high protein diet (45% protein, 30% carbohydrate) fed to Wistar rats had detrimental effects on the colonic microbiota compared to a normal protein diet (20% protein, 56% carbohydrate) [338]. *Streptococcus*, *E. coli/Shigella*, and *Enterococcus* increased by 5.36 fold, 54.9-fold, and 31.3 fold, respectively, on the high protein diet, whose abundances correlated positively with genes and metabolites associated with disease pathogenesis, including the metabolite cadaverine, which is derived from decarboxylation of lysine and in high amounts has been shown to induce oxidative stress and DNA damage [339]. Sulfate-reducing bacteria increased by 2.59 fold which correlated with increased sulfide. There was also an increase in spermine which has been shown to be extremely toxic in rats [340]. The following bacteria, which are generally regarded as being beneficial, decreased in abundance on the high protein diet, including the butyrate producer *F. prausnitzii* (by 3.5 fold), *Ruminococcus* (by 8.04 fold) which contains butyrate producers, and the mucin degrader *Akkermansia* (not detected in high protein diet group). *A. muciniphila* is considered to be a health promoting bacterium [341,342,343,344,345,346]. On the high protein diet, butyrate decreased by 2.16 fold. This diet was also associated with a decrease in genes involved in innate immunity, O-linked glycosylation of mucin, and oxidative phosphorylation. 

Protein source or type has also been shown to impact gut microbiota composition, given that the amino acid composition differs between types. A 14-day feeding trial in rats fed either protein from soy, pork, beef, chicken, fish and casein, (the latter served as a control) revealed changes by day 2 particularly between red meat (pork and beef) and white meat (fish and chicken). Principal component analysis revealed distinct microbiota on days 7 and 14, whereby the soy protein group was separate from the meat and casein groups [347]. In another similar study, soy protein was associated with increased faecal SCFAs in rats compared to rats fed white meat, red meat or casein [348]. The soy group also had a higher relative abundance of *Bacteroides* and *Prevotella* which are the major propionate and other SCFA producers [349]. *Lactobacillus*, a genus known for its beneficial effects was increased in the gut microbiota by ingestion of meat proteins. In this study, the diets clustered into two subgroups at the phylum level, the ‘meat class’ and the ‘non-meat class.’ In another study, soy fed hamsters were found to have a more consistently diverse microbiota in the small and lower intestine compared to hamsters fed milk protein isolate and the largest differences were found within the Bacteriodetes phylum [350]. Indeed, the milk protein isolate fed group had a greater relative abundance of *Bacteroidaceae* and *Porphyromonadaceae* compared to the soy fed groups. Those fed partially hydrolyzed soy protein isolate revealed a bloom of *Bifidobacteriaceae* across most intestinal sections and a higher proportion of *Clostridiales* spp. in the cecum compared to the milk protein isolate fed group. Bifidobacteria are well known beneficial microbes and commensal members of *Clostridiales* are now known to be involved in maintenance of overall gut function [351]. The milk protein isolate fed group had a greater abundance of *Erysipelotrichacaea* in ileum and faecal samples which has been associated with dyslipidemic phenotypes in humans [352] and hamsters [353]. Plasma lipids were significantly reduced in the soy fed hamsters compared to those that received milk protein isolate, owing at least in part to the gut microbiota changes induced by soy protein [350]. Under a high fat diet regime, soybean protein isolate reduced high fat diet induced weight gain and adipose tissue mass accumulation and attenuated hepatic steatosis in mice which was not observed for dairy protein [354]. An increased cecal bile acid pool was observed in the soybean group with an elevated secondary: primary bile acid ratio, along with an enhancement in the secretion of GLP-1. This was accompanied by an expansion of taxa proposed to be involved in bile acid biotransformation. These effects were abolished in germ free mice suggesting that the metabolic benefits of soy protein are due to the microbiota alterations which bring about increased bile acid transformation and secretion of GLP-1. Likewise, tartary buckwheat protein proved capable of preventing dyslipidemia in mice fed a high fat diet which was not observed for casein [355]. The buck wheat protein inhibited the growth of *E. coli* and promoted the growth of *Lactobacillus*, *Enterococcus* and *Bifidobacterium*, the latter of which was closely related to plasma lipids. Excretion of total bile acids and SCFAs were significantly increased in faeces from mice fed buckwheat. Mungbean protein also proved superior to casein in reducing high fat diet induced weight gain in mice [356]. The mungbean protein caused elevated secretion of GLP-1, an increase in the cecal and faecal bile acid pool with a dramatically elevated secondary: primary bile acid ratio; effects which were abolished in germ free mice. In terms of the gut microbiota, mungbean consumption as part of the high fat diet resulted in an expansion of *Ruminococcaceae*, a family known to have BSH activity [357] and caused an increase in taxa belonging to the Bacteroidetes phylum and a decrease in the abundance of Firmicutes [356]. Whey protein extract has been shown to increase diversity, *Bifidobacterium* and *Lactobacillus* populations while decreasing *Bacteroides* and *Clostridia* [358,359]. Similarly, pea protein extract has been shown to increase diversity, *Bifidobacterium* and *Lactobacillus* populations [360]. These studies clearly suggest that plant-derived proteins are superior to animal-derived proteins for promoting a beneficial microbiota with positive effects on host metabolism. 

Interestingly in humans, a three-week isocaloric supplementation with casein, soy protein or maltodextrin, which served as control, had no impact on the gut microbiota but altered bacterial metabolite production [361]. Compared to maltodextrin, both soy protein and casein caused a decrease in faecal butyrate which the authors suggest may be due to the combination of an increase in protein intake and a decrease in indigestible carbohydrate intake. Amino acid metabolites increased on the high protein diet owing to protein degradation by the gut microbiota. These metabolites included the faecal concentration of 2-methylbutyrate and the urinary concentration of the host-microbiota co-metabolites phenylacetylglutamine and indoxyl sulfate. Casein specifically increased p-cresol sulfate. Transcriptome analysis of rectal biopsies revealed changes in gene expression associated with mucosal homeostasis maintenance on the two protein diets, albeit the transcriptome profiles differed between the two, which the authors suggest is due to exposure to different bacterial metabolites resulting from the different proteins. However, mucosal inflammation was not induced on these diets and faecal water cytotoxicity was not altered either. The authors suggest that high protein diets should be considered with some caution given the changes observed in gene expression in the rectal mucosa and that protein source must also be taken into account. More recently, a low protein diet (0.6 g/kg/day) for six months was reported to reduce serum uremic toxin levels, including p-cresol sulfate, in non-dialysis, chronic kidney disease patients [362]. DGGE indicated a change in the intestinal microbiota profile. The low protein diet was also associated with an improvement in renal function and a reduction in total and LDL cholesterol. A position paper by the MyNewGut Study group analysed PubMed-referenced publications involving human intervention studies to clarify beneficial versus deleterious effects of high protein diets on metabolic and gut related health parameters, including interactions with the gut microbiota [363]. The study concluded that high protein diets are generally associated with decreased body weight and improvement in blood metabolic parameters but they also modify various bacterial metabolites and co-metabolites in faecal and urinary contents. The effects on the gut microbiota were heterogeneous depending on the type of dietary intervention. The effects of high protein diets on the gut microbiota were dependent on protein source (plant versus animal) and this should be considered for future investigations and some caution should be exercised around high protein diets, particularly as long-term or recurrent dietary practices. Note: Processing of protein, including thermal processing, and its impact on protein function, including modulation of the microbiome is not fully understood and requires further investigation. 

### 7.3. Carbohydrates

The quantity of dietary carbohydrates entering the colon each day has been estimated at approximately 40 g [364]. We have already seen that a carbohydrate-rich diet (fibre and plant-derived polysaccharides) results in a microbiota which is dominated by the *Prevotella* enterotype [33] and is enriched in polysaccharide degrading members with increased SCFA production [180,365,366,367,368,369]. However, dietary carbohydrates which enter the colon can be categorised as resistant starch, non-starch polysaccharides, oligosaccharides, as well as some di- and monosaccharides [370], and as with the other macronutrients, amount and type of carbohydrate has an impact on the gut microbiota. Resistant starch is an important non-digestible carbohydrate, of which there are four types, (RS1 to RS4) [371,372]. A diet high in type 3 resistant starch for 10 weeks was found to stimulate the growth of Firmicutes bacteria related to *R. bromii* and *Eub. rectales* in human volunteers [181], both of which possess amylolytic activity [373,374], and *Eub. rectales* members are major butyrate producers [375]. *Bifidobacterium* spp. were found to increase dramatically in one volunteer in response to the resistant starch used in this diet. In contrast, non-starch polysaccharides (wheat bran) presented little evidence of changes to the faecal microbiota, which the authors state may be attributable to the smaller increase achieved for non-starch polysaccharides (1.5 fold) compared to a 4.8 fold increase in resistant starch [181]. Interestingly, of the 14 volunteers, >60% of the resistant starch remained undigested in two compared to <4% in the remaining 12 volunteers highlighting the significance of the initial microbiota composition. Martínez et al. [376] revealed that resistant starch types 3 and 4 differentially affected the microbiota in human volunteers following consumption of each for three weeks in a double-blind cross-over study. In this case, resistant starch type 4 induced changes at the phylum level resulting in a decrease in Firmicutes and an increase in Actinobacteria and Bacteroidetes. At the species level, resistant starch type 4 increased *Bif. adolescentis* and *Para. distasonis*. In three subjects, resistant starch type 4 induced a 10-fold increase in bifidobacteria. Resistant starch type 2 significantly increased *R. bromii* and *Eub. rectales* compared to resistant starch type 4. Importantly, the responses to the resistant starches and their magnitudes varied between individuals but were reversible and tightly associated with resistant starch consumption. This interindividual variation was also reported by Venkataraman et al. [377]. In this case, resistant starch type 2 increased faecal butyrate in a cohort of 20 young healthy individuals but responses varied widely such that individuals could be categorised into three groups based on butyrate levels before and during resistant starch consumption: enhanced, high, and low. In the enhanced group, faecal butyrate increased by an average of 67% which coincided with a dramatic increase in the relative abundance of resistant starch-degrading organisms, including *Bif. adolescentis* or *R. bromii* in most individuals (increasing from 2% to 9%) and *Eub. rectales* in almost half of these individuals. In the high group, levels of butyrate remained the same before and during resistant starch consumption. In this group, resistant starch-degrading microorganisms increased in abundance, suggesting the resistant starch was degraded, but there was no concomitant increase in butyrate levels which the authors suggest may be due to these individuals experiencing a plateau effect owing to the fact that they are performing well in terms of butyrate production. In the low group, butyrate levels were low before resistant starch consumption began and did not improve during the trial. The resistant starch degrading bacteria in the low group did not increase from an initial abundance of ~1.5% suggesting their microbiota did not degrade the resistant starch which the authors state may be due to the presence of antagonistic microbes or lack of synergistic microbes. More recently, Vital et al. [378] reported that distinct parts of the microbiota work together to degrade resistant starch (type 2 in this study) and successively form health-promoting end products. In this case, resistant starch degradation was governed by Firmicutes whereby *R. bromii* degradation of the resistant starch provided fermentation substrates and increased acetate concentrations to support the growth of major butyrate producers. H_2_-scavenging sulphite reducers and acetogens also increased. Again, the individual responses varied with the observed pattern reported for seven of the twelve participants, while four showed mixed responses and one individual remained unresponsive. 

Using an *in vitro* approach based on isothermal microcalorimetry in conjunction with Illumina Miseq sequencing of the faecal microbiota and metabolite profiling, Adamberg et al. [379] compared the impact of different oligo- and polysaccharides (galacto- and fructooligosaccharides, resistant starch, levan, inulin, arabinogalactan, xylan, pectin and chitin), as well as a glycoprotein mucin, representing an indigenous substrate, on the growth and metabolism of the faecal microbiota. Apart from chitin, more than 70% of all substrates were fermented by the faecal microbiota with total heat generation up to 8 J/mL. Overall, the different fibre types supported the growth of specific microbiota and different metabolic routes. Arabinogalactan stimulated the growth of several health-promoting genera, including *Bifidobacterium, Bacteroides, Coprococcus* and *Lachnoclostridium*. Propionic acid was enhanced by arabinogalactin, xylan and mucin but not by galacto-, fructooligosaccharides or inulin. Fermentation of mucin resulted in the generation of acetate, propionate and the highest amount of butyrate and supported the growth of species from the following genera, *Clostridium, Lachnoclostridium* and *Parabacteroides*. The authors suggest that only a diverse spectrum of glycans in the diet can contribute to a diverse microbiota. Chung et al. [380] determined the impact of growth substrate complexity and multiplicity on species diversity of the human colonic microbiota *in vitro* whereby anaerobic fermenters were supplied continuously with single carbohydrates (arabinoxylanoligosaccharide, pectin or inulin), or with a mix of all three, or a mix of all three plus resistant starch, β-glucan and galacto-mannon as energy sources. Over the first six days, inulin supported less microbial diversity than the other single carbohydrates, the mix of three, or the mix of six carbohydrates. While the communities did not differ substantially at phylum and family levels, marked differences were observed at species level. The overall results strongly suggest that strategies to increase microbial diversity should employ complex nondigestible substrates mixtures. 

Few studies have investigated the impact of high sugar containing diets on the microbiota. However, recently, Sen et al. [381] investigated the impact of high fat/high sugar (sucrose), low fat/high sugar and low fat/low sugar diets on Sprague-Dawley rats. As expected, both the high fat/high sugar and low fat/high sugar diets for four weeks caused significant increases in body weight and body fat compared to the low fat/low sugar diet. The high sugar and high fat diets resulted in gut microbiota dysbiosis which was characterised by an overall decrease in microbial diversity, a bloom in clostridia and bacilli and a marked decrease in *Lactobacillus* species. There was also an increase in the Firmicutes: Bacteriodetes ratio. Specifically, the low fat/high sugar diet caused an increase in two *Proteobacteria* members, namely *Sutterella* and *Bilophila*. The high sugar diet also induced gut inflammation and triggered a remodelling of the gut brain axis [381]. In mice, a high sugar (sucrose) diet increased Clostridiales belonging to the Firmicutes phylum which was also observed for the high fat diet [382]. The high sugar diet also caused a decrease in the order Bacteroidales which belongs to Bacteroidetes. Lactobacillales was also significantly increased on the high sugar diet. Thus, overall the high sugar diet altered more bacterial orders and genera than the high fat diet. The increase in Clostridiales and decrease in Bacteroidales on the high sugar diet was related to poor cognitive flexibility. Di Luccia et al. [383] reported that a high fructose diet induced markers of metabolic syndrome, inflammation and oxidative stress in rats, but these were significantly reduced when the animals were treated with antibiotics or faecal samples from control rats fed a standard diet. Since the number of members from the genera *Coprococcus* and *Ruminococcus* were increased by the high fructose diet but reduced by antibiotics and faecal transplant, the results suggest a correlation between their abundance and metabolic syndrome.

## 8. Conclusions

The link between nutrition, the gut microbiota and health has been expounded in the multitude of studies published in recent years. Long-term dietary habits appear to have the most profound influence on the quality of the gut microbiota and hence its efficacy to the human body. In this regard, healthy eating patterns with adequate fruits and vegetables, ensuring a rich source of dietary fibre, along with healthy fats (MUFAs and PUFAs) and a trend towards more plant-derived proteins should help promote gut microbiota diversity and functionality enabling it to effectively serve its host. But as countries become more industrialised and range of choice expands, including foods available for consumption, consumers tend to eat for taste-satisfaction and/or convenience at the cost of nutritional value in many instances. Furthermore, not everyone responds effectively to dietary interventions aimed at improving health. The consequence of this has been an alarming surge in non-communicable diseases, including cancer and metabolic-related diseases. Understanding the specific role of the gut microbiota in the diet-health sequence has enabled scientists and nutritionists to further comprehend how diet specifically impacts health at the individual level and why dietary interventions do not always serve everyone equally. Thus, modification of the gut microbiota through diet, probiotics and prebiotics may offer viable opportunities for preventing many of these ‘Western-associated’ diseases, particularly in the case of metabolic-related diseases for which microbiota quality and quantity can be an essential aspect. Indeed, Part II of this review looks at the efficacy of such interventions in terms of over/under nutrition. We also explore opportunities for optimising health at different life stages (elderly in nursing homes, during pregnancy, physically active individuals and those in high stress environments) through improved diet and interventions. In this regard, microbiome testing at the individual level has a significant role to play in interpreting how an individual responds to dietary components and what interventions should be undertaken to improve those responses for a healthier outcome, thus underpinning the very goal of precision nutrition. In Part II of this review, we also look at specific examples of how the microbiota is already being used as a biomarker to predict responsiveness to specific dietary constituents and highlight further research which is necessary to make precision nutrition through the microbiome a reality.

## Figures and Tables

**Figure 1 nutrients-11-00923-f001:**
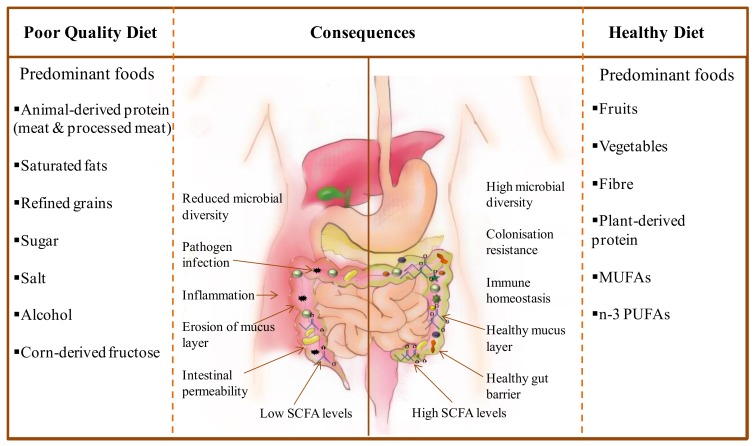
Comparison of consequences of poor-quality diet versus a healthy diet on the gut and gut microbiota (MUFAs = monounsaturated fatty acids; PUFAs = polyunsaturated fatty acids).

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
