# Peer review of "Precision Nutrition and the Microbiome, Part I: Current State of the Science"

_nutrients, 2019, doi:10.3390/nu11040923_

Round 1
Reviewer 1 Report
A well written review on nutrition and the microbiome, but overall the manuscript is just so long which makes it difficult to read. Have the authors thought about splitting this manuscript into a series of manuscripts to publish as a series?
Otherwise the manuscript is very well written and referenced.
Author Response
Response: We agree with Reviewer 1 that the review is too long; therefore we have split the review into two separate articles where Part 1 discusses the state of the science in terms of the gut microbiome and nutrition, entitled, Precision Nutrition and the Microbiome, Part I: Current State of the Science and Part II will focus on the potential use of the microbiome as a means to help prevent disease and promote health, entitled, Precision Nutrition and the Microbiome, Part II: Potential Opportunities and Pathways to Commercialisation.
Reviewer 2 Report
This review is well written and contains a vast amount of information but it's not all relevant to the topic of the paper. When reading the paper you get the impression its a literature review from a PhD thesis and therefore the review lacks focus and parts of the review is simplified as its written for a non-scientific reader. The length of review is also excessive due to the lack of focus.
I acknowledge that it's important to give an overview of where we are at in terms of understanding the interaction between the microbiome, diet and the host, but this takes up the majority of the review (30 pages). It would be beneficial to summaries this information in a table. In my opinion the review should focus on how you can change the microbiome as that would create a coherent story around precision nutrition and the microbiome.
The latter part of the review is of interest for a non-scientific reader not familiar with these limitations but to do it justice its almost an review in itself and range of other factors not mentioned in the paper plays a significant role, i.e. how the sequence library is build. I would therefore recommend to either exclude this part or congest it to a paragraph highlighting you are aware of these limitations and alert to it.
Author Response
Response: We have acknowledged that the review is too long and have generated two manuscripts (see response to Reviewer 1). While Part II is in the process of being completed, we feel that each review is now much more focused (Part I: Current State of the Science and Part II: Potential Opportunities and Pathways to Commercialisation) but together provide a coherent narrative for the reader.
We disagree with the suggestion of providing a table in relation to understanding the interaction between the microbiome, diet and the host in place of the text as we will not be able to condense the content into a table and provide a decent overview of the current state of science. We feel that presenting this information in a table with references does not add value to the reader/audience as they will have to chase down additional papers to get the overview that we have already summarised on their behalf.
In relation to the latter part of the review being of interest to a non-scientific reader, we disagree. This part of the submitted review is now in Part II. Companies currently operating or looking to operate in this space will gain value from this information. It brings attention to some key factors that need to be considered whilst at the same time highlighting the commercial potential. Excluding or congesting this section will remove one of the most valuable aspects of the review that is often neglected. Commercialisation of new technologies in the early stages of development is extremely important as it provides an opportunity to understand consumer engagement and to collect consumer data that can accelerate research. Indeed, Reviewer 3 stated that sections 9 and 10 were the most novel and they were well discussed and reviewed.
Reviewer 3 Report
This is a well-written and very comprehensive review about nutrition-gut microbiota interactions. The extension of this review is maybe quite long and more appropriate for a book chapter so make sure it fit with Nutrients guidelines for this king of manuscript format.
Although the field of the gut microbiota has been extensively discussed in the literature in the last years from many different points of views involving different aspects such as nutrition, not many publications have discussed the potential of the gut microbiota in precision medicine and in precision microbiomics. In this regard, this manuscript contribution to this field is interesting and needed.
INTRODUCTION
- Line 35. A relatively recent study from Sender et al published in (Cell) (DOI:https://doi.org/10.1016/j.cell.2016.01.013) discussed that the total number of bacteria has been outnumbered by approx.. ten times. Applying more accurate calculations, they concluded that this number is closed to 1013 than to 1014. I think it is important to update this information in current publications referring to this, as it seems that human cells:bacteria cells ratio is closed to 1:1 rather than to 1:10 as it has been published in many papers.
- Line 69: include PMID 29471863
- In section 2: include https://doi.org/10.1017/S0029665114000627 “Intestinal microbiota during early life – impact on health and disease”.
- Line 118 and forward: Diet is a very important factor shaping the gut microbiota. Authors should also indicate if in the studies that investigated the microbiota at different age (children, pre- and adolescents, healthy adults, etc) the populations were from similar regions and/or similar diets. In other words, were these studies comparable from a diet point of view?
- Line 280: Butyrate has also been shown to play an important role in brain function https://doi.org/10.1016/j.neuint.2016.06.011
- Line 284: Propionate has also been shown to reduce cancer https://www.nature.com/articles/bjc2012409
- Line 289-294: SCFA like acetate has also been shown to impact metabolism in a negative way promoting metabolic syndrome: https://www.nature.com/articles/nature18309
- 302-313: gut microbiota also inhibits bile acid synthesis in the liver by alleviating FXR inhibition in the ileum https://www.sciencedirect.com/science/article/pii/S1550413113000119
- 314-334: cita also https://www.nature.com/articles/s41564-018-0307-3?fbclid=IwAR2Pm9kivBhG5Eeh0dnOxRQUrZha_Lbpv9U6XtoSc8KkrvHMmt3I-CkndFU
- 335-353: cite also https://www.pnas.org/content/115/25/6458.short
- 409-426: cite also https://journals.plos.org/plosone/article?id=10.1371/journal.pone.0164036, https://www.ncbi.nlm.nih.gov/pmc/articles/PMC6173059/, https://www.sciencedirect.com/science/article/pii/S0966842X18300180
- 651: missing reference for link MS-diabetes
- 694-697: as in my comment above, acetate promotes metabolic syndrome: https://www.nature.com/articles/nature18309, so this paper may also be cited here
- In section 7 include references as: https://www.sciencedirect.com/science/article/pii/S193131281830266X, https://www.ncbi.nlm.nih.gov/pmc/articles/PMC6020729/,
- mention the particular case of ketogenic diet (low carbs and protein and high fat diet) which has shown different beneficial effects and to affect gut microbiota composition https://www.sciencedirect.com/science/article/pii/S0092867418305208 https://www.ncbi.nlm.nih.gov/pmc/articles/PMC5692397/ https://www.nature.com/articles/s41598-018-25190-5/
- section 8: include this paper in order to discuss as well that probiotic-mediated beneficial effects may not require the viability of the bacteria https://www.ncbi.nlm.nih.gov/pubmed/27892954
- sections 9 and 10 are the most novel and they are well discussed and reviewed.
Author Response
This is a well-written and very comprehensive review about nutrition-gut microbiota interactions. The extension of this review is maybe quite long and more appropriate for a book chapter so make sure it fit with Nutrients guidelines for this king of manuscript format.
Although the field of the gut microbiota has been extensively discussed in the literature in the last years from many different points of views involving different aspects such as nutrition, not many publications have discussed the potential of the gut microbiota in precision medicine and in precision
1. Line 35. A relatively recent study from Sender et al published in (Cell) (DOI:https://doi.org/10.1016/j.cell.2016.01.013) discussed that the total number of bacteria has been outnumbered by approx.. ten times. Applying more accurate calculations, they concluded that this number is closed to 1013 than to 1014. I think it is important to update this information in current publications referring to this, as it seems that human cells:bacteria cells ratio is closed to 1:1 rather than to 1:10 as it has been published in many papers.
Response: We agree with the Reviewer and have now changed the number to 1013 and referenced Sender et al., 2016 (line 40).
Line 69: include PMID 29471863
Response: The above reference has now been included (line 74).
In section 2: include https://doi.org/10.1017/S0029665114000627 “Intestinal microbiota during early life – impact on health and disease”.
Response:We have included the above reference in Section 2 (line 111).
Line 118 and forward: Diet is a very important factor shaping the gut microbiota. Authors should also indicate if in the studies that investigated the microbiota at different age (children, pre- and adolescents, healthy adults, etc) the populations were from similar regions and/or similar diets. In other words, were these studies comparable from a diet point of view?
Response: In order to address this comment, we have provided nationality of subjects or type of diet consumed where such information is available.
Line 280: Butyrate has also been shown to play an important role in brain function https://doi.org/10.1016/j.neuint.2016.06.011
Response: We have now included the above reference and referred to butyrate in terms of brain function (line 289).
Line 284: Propionate has also been shown to reduce cancer https://www.nature.com/articles/bjc2012409
Response: We have included the above reference as follows (line 295): ‘Propionate derived from the gut microbiota has also been shown to reduce cancer cell proliferation in the liver (Bindels et al., 2012).’
Line 289-294: SCFA like acetate has also been shown to impact metabolism in a negative way promoting metabolic syndrome: https://www.nature.com/articles/nature18309
Response: We have included the above reference as follows (line 305): “As an example, increased acetate production from an altered gut microbiota in a rodent model was shown to promote metabolic syndrome (Perry et al., 2016).”
302-313: gut microbiota also inhibits bile acid synthesis in the liver by alleviating FXR inhibition in the ileum https://www.sciencedirect.com/science/article/pii/S1550413113000119
Response: We have included the above reference as follows (line 325): “The gut microbiota has also been shown to inhibit bile acid synthesis in the liver by alleviation of FXR inhibition in the ileum (Sayin et al., 2013).”
314-334: cita also https://www.nature.com/articles/s41564-018-0307-3?fbclid=IwAR2Pm9kivBhG5Eeh0dnOxRQUrZha_Lbpv9U6XtoSc8KkrvHMmt3I-CkndFU
Response: We have included the above reference as follows (line 337): “More recently, a co-culture experiement revealed that GABA produced by Bacteroides fragilis was essential for the growth of a gut isolate termed KLE1738 which is believed to be an unreported bacterial genus (Strandwitz et al., 2019). This led to the isolation of a variety of GABA-producing bacteria and the Bacteriodes species in particular were found to produce large quantities of GABA. Furthermore, in the same study relative abundance levels of faecal Bacteriodes negatively correlated with brain signatures associated with depression in patients with major depressive disorder.”
335-353: cite also https://www.pnas.org/content/115/25/6458.short
Response: The above reference has now been cited (line 366).
409-426: cite also https://journals.plos.org/plosone/article?id=10.1371/journal.pone.0164036, https://www.ncbi.nlm.nih.gov/pmc/articles/PMC6173059/, https://www.sciencedirect.com/science/article/pii/S0966842X18300180
Response: The above 3 references have been cited in the appropriate section (lines 428, 429 and 446). Umu et al., 2016 (line 446) has been explained as follows: “Supplementing mice with bacteriocin-producing strains resulted in transient advantageuous changes such as inhibition of Staphylococcus by enterocins and Enterococcus by garvicin and promotion of LAB by sakacin, plantaricins and garvicin (Umu et al., 2016).”
651: missing reference for link MS-diabetes
Response: The appropriate reference (line 677) (Magkos F, Yannakoulia M, Chan JL, Mantzoros CS. 2009. Management of the metabolic syndrome and type 2 diabetes through lifestyle modification. Annual Review of Nutrition 29, 223-256.) has now been added.
694-697: as in my comment above, acetate promotes metabolic syndrome: https://www.nature.com/articles/nature18309, so this paper may also be cited here
Response: The reference “Perry et al., 2016” has also been cited here (line 721) as suggested.
In section 7 include references as: https://www.sciencedirect.com/science/article/pii/S193131281830266X, https://www.ncbi.nlm.nih.gov/pmc/articles/PMC6020729/,
Response: We have included one of the above references (Makki K, Deehan EC, Walter J, Bäckhed F. 2018. The impact of dietary fiber on gut microbiota in host health and disease. Cell Host and Microbe 23, 705-715) at line 1222.
However, we feel the other recommended reference does not fit with this section (Shondelmyer et al., 2018. Ancient Thali diet: Gut microbiota, immunity and health. Yale Journal of Biology and Medicine, 91, 177-184) given that it focuses largely on phytochemicals in the diet.
mention the particular case of ketogenic diet (low carbs and protein and high fat diet) which has shown different beneficial effects and to affect gut microbiota composition https://www.sciencedirect.com/science/article/pii/S0092867418305208 https://www.ncbi.nlm.nih.gov/pmc/articles/PMC5692397/ https://www.nature.com/articles/s41598-018-25190-5/
Response: As recommended by the Reviewer we have mentioned the ketogenic diet and cited the above references as follows (line 1091): “The ketogenic diet, defined as a high fat and low carbohydrate diet, has been effectively used as a therapeutic treatment for a number of neurological disorders including epilepsy, Alzheimer’s disease, Parkinson’s disease, depression, autism, traumatic brain injury and depression, as examples (Baranano et al., 2008). Recently, researchers have revealed that the protective effects of the ketogenic diet could be mediated via the gut microbiota based on studies in mice (Olson et al., 2018; Ma et al., 2018). The ketogenic diet administered by Ma et al., (2018) consisted of 75.1% fat composed of SFAs, MUFAs and PUFAs. Following 16 weeks on the diet, mice revealed several neurovascular enhancements with potential to reduce the risk of Alzheimer’s disease which could be associated with the observed gut microbiota changes which included an increase in beneficial bacteria including Akkermansia muciniphila and Lactobacillus and a reduction in pro-inflammatory microbes Desulfovibrio and Turicibacter. The gut microbiota changes observed by Olson et al., (2018) in mice receiving the ketogenic diet were found to be required for the protective effects of the diet against acute electrically induced seizures and spontaneous tonic-clonic seizures. In this case, Akkermansia and Parabacteriodes were significantly increased and enrichment of, and gnotobiotic colonisation with these microorganisms were capable of restoring seizure protection in germ-free mice or mice treated with antibiotics. Additionally, in a mouse model of glioma, mice fed the ketogenic diet had slightly increased survival compared to mice fed the control diet and showed significant differences in several key microorganisms (McFarland et al., 2017). The ketogenic diet generally includes fats of all chain lengths (Stafstrom and Bough, 2003) thus increases in blood ketone levels could be responsible for the observed gut microbiota changes (Ma et al., 2018).”
section 8: include this paper in order to discuss as well that probiotic-mediated beneficial effects may not require the viability of the bacteria https://www.ncbi.nlm.nih.gov/pubmed/27892954
Response: This section is now in Part II of the review. However, we agree with the Reviewer and will cite the above reference as suggested.
sections 9 and 10 are the most novel and they are well discussed and reviewed.
Round 2
Reviewer 2 Report
Thank you for revised manuscript and I pleased to see that you have chosen to split it into two manuscripts. I accept your arguments for keeping the text and not producing a table.